EMBO
Molecular Medicine

# Reelin-LRP8 signaling mediates brain dissemination of breast cancer cells via abluminal migration

Haofeng Huang [1,7], Min Zhang [1,7], Enyu Huang[1,7], Yongfang Zhao[2], Xiaoyu Li[1], Pu Qiu[3], Cairui Li[4], Jiahua Tao [5], Yuanqi Zhang[3], Lianxiang Luo[5], Guozhu Ning [1✉], Ceshi Chen[2] & Jingjing Zhang [1,6✉]

## Abstract

**Brain metastasis (BM) remains a significant challenge in breast cancer (BC) management. While conventional metastatic routes primarily involve hematogenous dissemination, emerging evidence suggests that BC cells can also migrate along the abluminal surface of blood vessels, bypassing the blood-brain barrier (BBB). To investigate this phenomenon, we established a zebrafish xenograft model utilizing GFP-labeled MDA-MB-231 cells, allowing real-time observation of BC cell migration along the posterior cerebral veins. Our findings revealed that LRP8, an apolipoprotein E receptor, is upregulated in BC patients with brain metastasis. Functional studies demonstrated that *LRP8* knockdown significantly inhibited proliferation, migration, and invasion of triple-negative breast cancer (TNBC) cells both in vitro and in vivo. Mechanistically, LRP8 promotes the activation of CDC42, enhancing filopodia formation and cell motility, a process influenced by the neuronal extracellular matrix protein, Reelin. Furthermore, we demonstrated the therapeutic potential of MEN 10207, a neurokinin-2 receptor antagonist, in inhibiting TNBC cell migration and suppressing BM formation in both zebrafish and mouse models. These findings provide novel insights into the mechanisms underlying extravascular brain dissemination of BC, highlighting the Reelin-LRP8-CDC42 axis as a potential therapeutic target for this devastating complication.**

**Keywords** Breast Cancer; Brain Metastasis; Xenografted Model; LRP8; Reelin
**Subject Category** Cancer

## Introduction

According to the Global Cancer Observation (GLOBOCAN), breast cancer (BC) is now the most prevalent cancer worldwide, affecting 2.3 million women and resulting in 665,000 deaths annually (Bray et al, 2024). Metastasis rates in BC have also increased, with over 20% of patients at risk of developing distant metastases (Cantalejo-Díaz et al, 2024). BC cells can spread from their primary site to organs such as the brain, bones, liver, and lungs, significantly raising mortality rates and complicating treatment (Farahani et al, 2023). Among metastatic cases, brain metastasis (BM) is the deadliest complication of BC and the incidences of breast cancer brain metastasis (BCBM) continue to rise (Chow et al, 2015). BM substantially impacts morbidity and mortality, and in female patients, BC is the most common primary cancer source for BM (Boire et al, 2020; Lin et al, 2004). Despite various treatment options, including surgery, chemotherapy, radiotherapy, and immunotherapy, the prognosis for BC with brain metastasis remains poor and its molecular mechanisms are not well understood (Corti et al, 2022). Improving the prognosis and treatment of BCBM thus remains a substantial challenge.

BCBM manifests in three types of brain metastasis: parenchymal metastasis (92% of cases), leptomeningeal metastasis (8%), and the rare choroid plexus metastasis (Wang et al, 2021). Traditionally, BCBM was thought to occur as BC cells infiltrate nearby vasculature, spread through the circulatory system, and cross the blood-brain barrier (BBB) to invade the central nervous system (CNS) (Wilhelm et al, 2013). However, recent findings suggest that BC cells in murine models can bypass the BBB by migrating along the abluminal surface of emissary veins to access the leptomeningeal space (Whiteley et al, 2024). This newly identified migration pathway offers an alternative route for BCBM but has received limited attention. Our study aims to establish a rapid, in vivo model to visualize the dynamic abluminal migration of brain-metastatic BC cells, as conventional murine xenograft models present challenges for tracking progression in real time (Gamble et al, 2021; Stoletov et al, 2007). A zebrafish xenograft model offers a promising alternative.

A recent report highlighted that glial-derived neurotrophic factor (GDNF), secreted by reactive CNS microglia, supports BC cell survival and proliferation in the leptomeninges, underscoring its role in BCBM progression (Whiteley et al, 2024). Another study found that neuropeptide substance P, produced by sensory neurons, could enhance BC proliferation and metastasis (Padmanaban et al, 2024).

[1]Zhanjiang Key Laboratory of Zebrafish Model for Development and Disease, Affiliated Hospital of Guangdong Medical University, 524001 Zhanjiang, China. [2]Yunnan Key Laboratory of Breast Cancer Precision Medicine, Academy of Biomedical Engineering, Kunming Medical University, 650000 Kunming, China. [3]Department of Breast Surgery, Affiliated Hospital of Guangdong Medical University, 524001 Zhanjiang, China. [4]Dali Bai Autonomous Prefecture People's Hospital (The Third Affiliated Hospital of Dali University), 671000 Dali, China. [5]The First Clinical College & School of Ocean and Tropical Medicine, Guangdong Medical University, 524021 Zhanjiang, China. [6]School of Medical Technology, Guangdong Medical University, 523808 Dongguan, China. [7]These authors contributed equally: Haofeng Huang, Min Zhang, Enyu Huang.
✉E-mail: ningguozhu@gdmu.edu.cn; jingjing.zhang@gdmu.edu.cn

These findings suggest that the brain microenvironment contributes to BC progression within the brain. Additionally, Reelin, a glycoprotein primarily located in GABAergic interneurons of the prefrontal cortex, hippocampus, and other brain regions, influences neuronal differentiation and migration via the low-density lipoprotein receptor-related protein 8 (LRP8), also known as apolipoprotein E receptor 2 (APOER2) (Impagnatiello et al, 1998; Reddy et al, 2011).

Emerging evidence suggests that Reelin and its receptor, LRP8, may play critical roles in BC progression. For instance, astrocytes modulate Reelin expression, which interacts with Her2 to drive BC cell proliferation and spheroid formation (Jandial et al, 2017). Furthermore, elevated LRP8 expression has been linked to poorer prognosis in BC patients (Lin et al, 2018). However, the mechanisms through which brain-derived Reelin promotes brain metastasis of BC via LRP8 receptors on BC cell membranes remain poorly understood. According to the "seed and soil" hypothesis, metastatic BC cells require a permissive microenvironment to colonize the brain (Langley and Fidler, 2011). This underscores the importance of investigating the Reelin-LRP8 pathway within the neuronal microenvironment as a potential avenue for therapeutic intervention. To this end, we employed molecular docking-based drug screening to identify small-molecule compounds targeting the Reelin-binding domain of LRP8, aiming to disrupt this signaling axis and inhibit brain dissemination of TNBC cells effectively. We then evaluated the therapeutic efficacy of MEN 10207 in zebrafish and nude mice xenograft models, screening out a promising candidate for the treatment of BCBM.

In this study, we leveraged a zebrafish xenograft model to visualize the real-time abluminal migration of brain-metastatic BC cells. This innovative model allowed us to explore the molecular pathways involved in BCBM, with a particular focus on the Reelin-LRP8 signaling axis. Additionally, we identified and validated a novel therapeutic candidate targeting this pathway, and demonstrated the anti-BCBM efficacy of MEN 10207 in zebrafish and mammal xenograft model, offering promising directions for future BCBM therapies.

## Results

### Abluminal migration of MDA-MB-231 cells in zebrafish

To investigate the abluminal migration of BC cells in vivo, we established a zebrafish xenograft model. As BC cells expression of integrin α6 is essential for abluminal migration (Whiteley et al, 2024) and integrin α6 is the second most abundant mRNA in the MDA-MB-231 cell line relative to other human BC cell lines (Fig. EV1), we injected MDA-MB-231 (GFP$^+$) cells into the perivitelline space of $Tg(kdrl:mCherry)$ zebrafish embryos, which express red fluorescent protein in the vasculature (Fig. 1A). Time-lapse imaging revealed that MDA-MB-231 cells migrated along the abluminal surface of blood vessels, predominantly the posterior cerebral vein (PCeV) (Fig. 1B), without entering the bloodstream (Fig. 1C,D; Movies EV1–3). By 72 h post-injection (hpi), a significant proportion of tumor cells had reached the brain parenchyma (Fig. 1E; Movie EV4). Approximately 60% of transplanted zebrafish embryos exhibited brain metastasis of MDA-MB-231 cells, demonstrating the efficiency of this abluminal

migration pathway. These findings provide strong evidence that human TNBC cells can efficiently migrate to the brain via an abluminal route. This zebrafish xenograft model offers a powerful tool to study the mechanisms underlying this process and to identify potential therapeutic targets.

### LRP8 is highly expressed in TNBC and predicts a high risk of brain metastasis

To identify genes associated with BCBM, we analyzed two GSE microarray datasets: GSE100534, which includes 16 breast tumor samples and 3 breast cancer brain metastases samples, and GSE52604, which consists of 10 fresh frozen non-neoplastic breast tissue samples and 35 fresh frozen breast brain metastasis samples. These datasets were used to identify potential brain metastasis-related genes, resulting in the discovery of 26 co-expressed differentially expressed genes (co-DEGs) through Venn diagram analysis (Fig. 2A). The heatmaps showed the upregulated genes in association with BCBM (Fig. 2B,C). The KM plotter databases (http://kmplot.com) indicated that high expression of LRP8, which is highly expressed in TNBC and predicts a high risk of brain metastasis, was correlated with a decreased likelihood of distant metastasis-free survival (DMFS) in BC patients (Fig. 2D). The Table EV1 also showed that LRP8 exhibited the highest hazard ratio (HR) among the 26 analyzed genes. Furthermore, the overall survival (OS) analysis indicated that BC patients with high levels of LRP8 expression experienced poorer survival outcomes (Fig. 2E). Besides, by accessing the UALCAN databases (http://ualcan.path. uab.edu), LRP8 expression is higher in BC patients than in normal individuals (Fig. 2F), and the expression in TNBC patients is higher than in other subclasses (Fig. 2G). Pathological sections from TNBC patients were analyzed to detect the LRP8 expression via IHC assay. LRP8 was highly expressed in TNBC primary tissues compared with paracancerous tissues (Fig. 2H,I). We subsequently assessed the protein levels of LRP8 in MCF-10A, MCF-7, BT549 and MDA-MB-231 cells, revealing that LRP8 expression is significantly elevated in TNBC cell lines compared with MCF-10A and MCF-7 cells (Fig. 2J,K). Additionally, TNBC patients have a heightened risk of developing BM compared with luminal BC patients and normal individuals (Franchino et al, 2018). Thus, these results suggest that LRP8 overexpression in TNBC is associated with an increased risk of brain metastasis.

### *LRP8* knockdown inhibits proliferation, migration and invasion of TNBC cells

To investigate the role of LRP8 in TNBC cell proliferation, we first knocked down *LRP8* expression in MDA-MB-231 and BT549 cells using specific shRNAs. Western blot analysis confirmed efficient LRP8 knockdown in cells transfected with *sh-LRP8-1$^\#$* and *sh-LRP8-2$^\#$* compared to the *sh-ctrl* group (Figs. 3A,B and EV2A,B). Subsequently, CCK-8 assays demonstrated a significant reduction in cell viability in *LRP8*-depleted cells (Figs. 3C and EV2C). To further assess the impact of LRP8 on cell proliferation, we performed colony formation assays. *LRP8* knockdown cells exhibited a marked decrease in colony formation ability compared to control cells (Figs. 3D,H and EV2D,E). Immunofluorescence staining for KI67, a proliferation marker, revealed a significant reduction in cell proliferation in *LRP8*-depleted cells (Fig. EV2F–I).

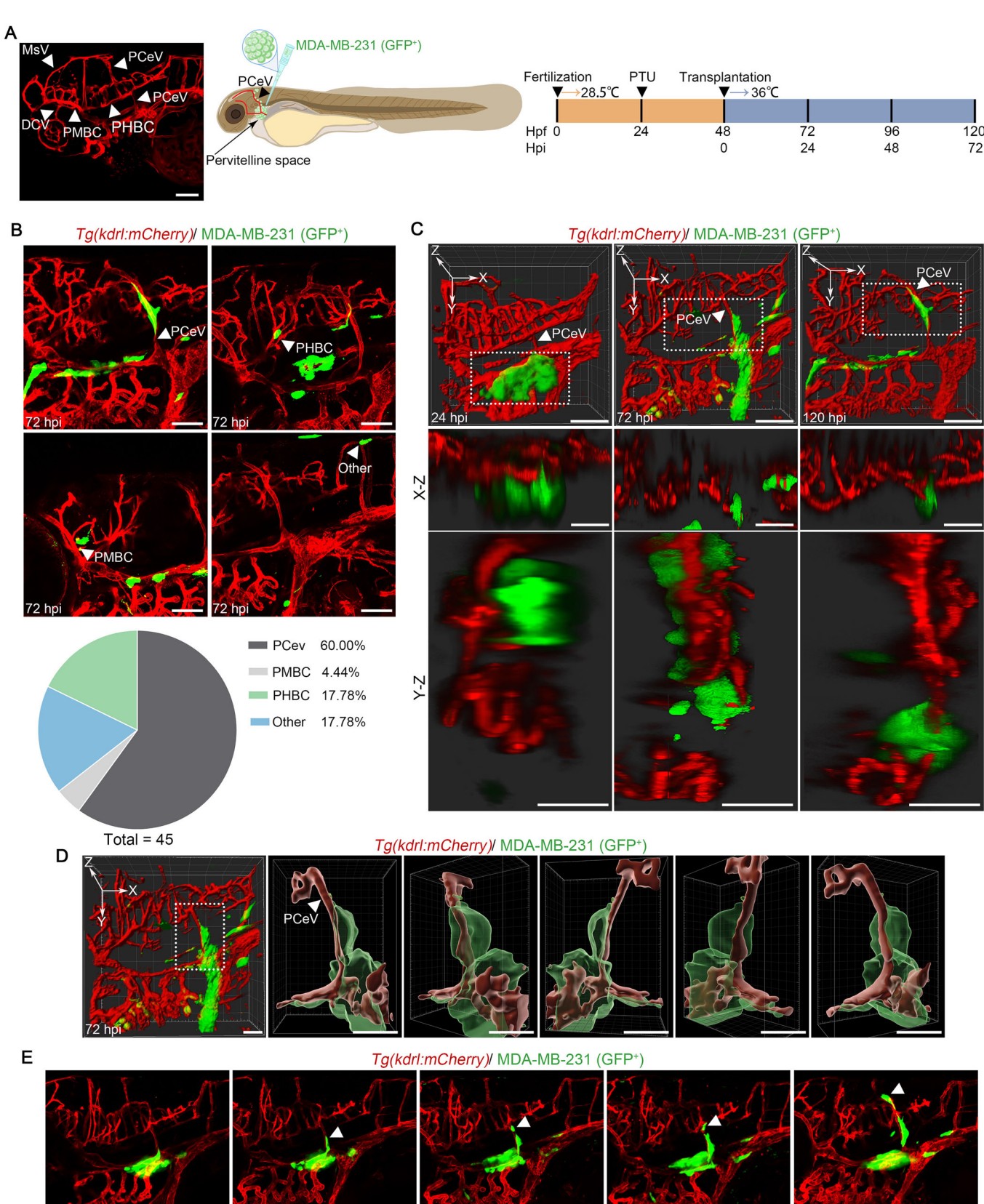

◄ **Figure 1. Establishment of a zebrafish xenograft model for analyzing BCBM.**

(A) Confocal image illustrates the visualization of the circulatory system in *Tg(kdrl:mCherry)* zebrafish at 4 dpf and schematic representation illustrates the zebrafish xenograft model using MDA-MB-231 (GFP$^+$) cells. Scale bar: 100 μm. (B) Top: Following the implantation of MDA-MB-231 (GFP$^+$) cells into zebrafish, representative confocal microscopy images show the localization of MDA-MB-231 cells within various blood vessels. MDA-MB-231 cells are displayed in green, and the vasculature is shown in red. Scale bar: 100 μm. Bottom: A pie chart illustrates the distribution ratio of MDA-MB-231 cells across distinct blood vessels, based on an analysis of $n = 45$ zebrafish. (C, D) Three-dimensional imaging (scale bar: 100 μm) and reconstruction (scale bar: 50 μm) were performed to visualize the interaction between transplanted MDA-MB-231 cells and the PCeV in zebrafish. The MDA-MB-231 cells are shown in green, and the vasculature is depicted in red. Scale bar: 100 μm. (E) Representative confocal microscopy images of larvae injected with MDA-MB-231 cells at 24, 36, 48, 60, and 72 hpi. MDA-MB-231 cells are displayed in green, and the vasculature is shown in red. The white arrow heads indicated the disseminated cancer cells. Scale bar: 100 μm. PHBC primordial hindbrain channel, PMBC primordial midbrain channel, DCV dorsal ciliary vein, MsV mesencephalic vein. Source data are available online for this figure.

To validate these findings in vivo, we utilized a zebrafish xenograft model. MDA-MB-231 (GFP$^+$) cells were injected into the perivitelline space of zebrafish larvae, and KI67 staining was performed at 72 hpi. A lower relative GFP fluorescence intensity was observed in zebrafish implanted with *LRP8* knockdown cells, suggesting reduced tumor growth (Fig. 3E,I). Additionally, zebrafish implanted with *LRP8* knockdown cells exhibited reduced KI67 staining compared to control cells, indicating decreased proliferation in vivo (Fig. 3F,J). These findings demonstrate that LRP8 knockdown effectively inhibits TNBC cell proliferation both in vitro and in vivo.

The upregulation of LRP8 in BCBM tissue suggests its potential involvement in BCBM progression (Fig. 2B,C). To investigate this hypothesis, we assessed the impact of *LRP8* depletion on the migratory and invasive abilities of TNBC cells. Wound-healing and transwell migration assays revealed that *LRP8* knockdown significantly impaired the migratory capacity of TNBC cells (Figs. 3G,K and EV3A–F). Similarly, transwell invasion assays demonstrated that *LRP8* knockdown significantly reduced the invasive potential of these cells (Figs. 3G,L and EV3E,G). To further validate these findings in vivo, we utilized a zebrafish xenograft model. MDA-MB-231 (GFP$^+$) cells transfected with *sh-ctrl* or *sh-LRP8-2#* were injected into the perivitelline space of *Tg(kdrl:mCherry)* zebrafish. *LRP8* knockdown significantly reduced the abluminal migration of brain metastatic BC cells (Fig. 3M). Overall, these results indicate that LRP8 plays a crucial role in promoting the proliferation, migration and invasion of TNBC cells, both in vitro and in vivo.

## Loss of *LRP8* reduces GTP-CDC42 expression, leading to inhibition of filopodia formation

To elucidate the underlying mechanism of LRP8-mediated migration and invasion, we performed RNA-seq analysis on MDA-MB-231 cells transfected with *sh-ctrl* and *sh-LRP8-2#*. Gene set enrichment analysis revealed that several downregulated genes in *LRP8*-depleted cells were involved in p75NTR-mediated activation of RAC and CDC42 via guanine nucleotide exchange factors (GEFs) (Fig. 4A). Heatmap analysis further confirmed the downregulation of multiple GEFs in *LRP8*-deficient cells (Fig. 4B). Protein-protein interaction (PPI) network analysis using STRING indicated that CDC42 interacts with these GEFs (Fig. 4C). GEFs activate Rho GTPases, such as CDC42, by catalyzing the exchange of GDP for GTP (Fig. 4D) (Bekere et al, 2021). The qRT-PCR analysis validated the downregulation of key GEFs, including ARHGEF38, ARHGEF37, ARHGEF16, and FGD4 (Fig. 4E). The active GTP-bound form of CDC42 plays a pivotal role in filopodia

formation (Scharler et al, 2022), which is crucial for cell migration and invasion (Bischoff et al, 2021; Jacquemet et al, 2015). To investigate the impact of LRP8 on CDC42 activation, we measured the levels of GTP-bound CDC42 in MDA-MB-231 cells. *LRP8* knockdown resulted in a significant decrease in GTP-CDC42 levels (Fig. 4F,G). To assess the functional consequences of reduced CDC42 activation, we examined filopodia formation in MDA-MB-231 cells. Phalloidin staining revealed a significant reduction in the number and length of filopodia in *LRP8*-depleted cells (Fig. 4H–J). These findings suggest that LRP8 regulates the expression of GEFs, leading to increased CDC42 activation and subsequent filopodia formation, which is essential for TNBC cell migration and invasion.

## Reelin-LRP8 signaling pathway activates CDC42 to regulate migration and invasion of MDA-MB-231 cells

Reelin, an upstream regulator of LRP8, is known to influence neuronal differentiation and migration through its interaction with LRP8 (Impagnatiello et al, 1998). To investigate the potential role of Reelin in regulating TNBC cell behavior, we performed in vitro functional assays. While Reelin treatment did not affect LRP8 expression or cell proliferation in both *sh-ctrl* and *sh-LRP8-2#* MDA-MB-231 cells (Fig. EV4A–F), it significantly enhanced cell migration and invasion in *sh-ctrl* cells, but not in *sh-LRP8-2#* cells (Figs. 5A–C and EV5G,H). These findings suggest that Reelin exerts its effects on cell migration and invasion through LRP8. To further explore the underlying mechanism, we examined the activation of CDC42 following Reelin treatment. Reelin treatment increased the levels of GTP-bound CDC42 in *sh-ctrl* cells, but not in *sh-LRP8-2#* cells (Fig. 5D,E). These results indicate that Reelin activates CDC42 through LRP8. To validate these findings in vivo, we utilized a zebrafish xenograft model. Treatment with an anti-Reelin monoclonal antibody significantly reduced the brain metastatic ability of MDA-MB-231 cells compared to an anti-IgG antibody control (Fig. 5F,G). These results suggest that the Reelin-LRP8 signaling pathway activates CDC42, which in turn promotes the migration and invasion of TNBC cells.

To derive the brain higher metastatic cell lines and evaluate the role of Reelin-LRP8-CDC42 signaling in cell lines with enhanced brain metastatic ability, we used this zebrafish xenograft model to isolate populations of MDA-MB-231 cells that migrated into the brain (Fig. 5H). We then assessed *LRP8* expression across multiple cell generations and found the mRNA of *LRP8* expression has a significant increase in the sixth-generation of brain metastatic (BM6) cell line (Fig. 5I). To gain deeper insights into the molecular changes underlying increased brain metastatic potential, RNA-Seq

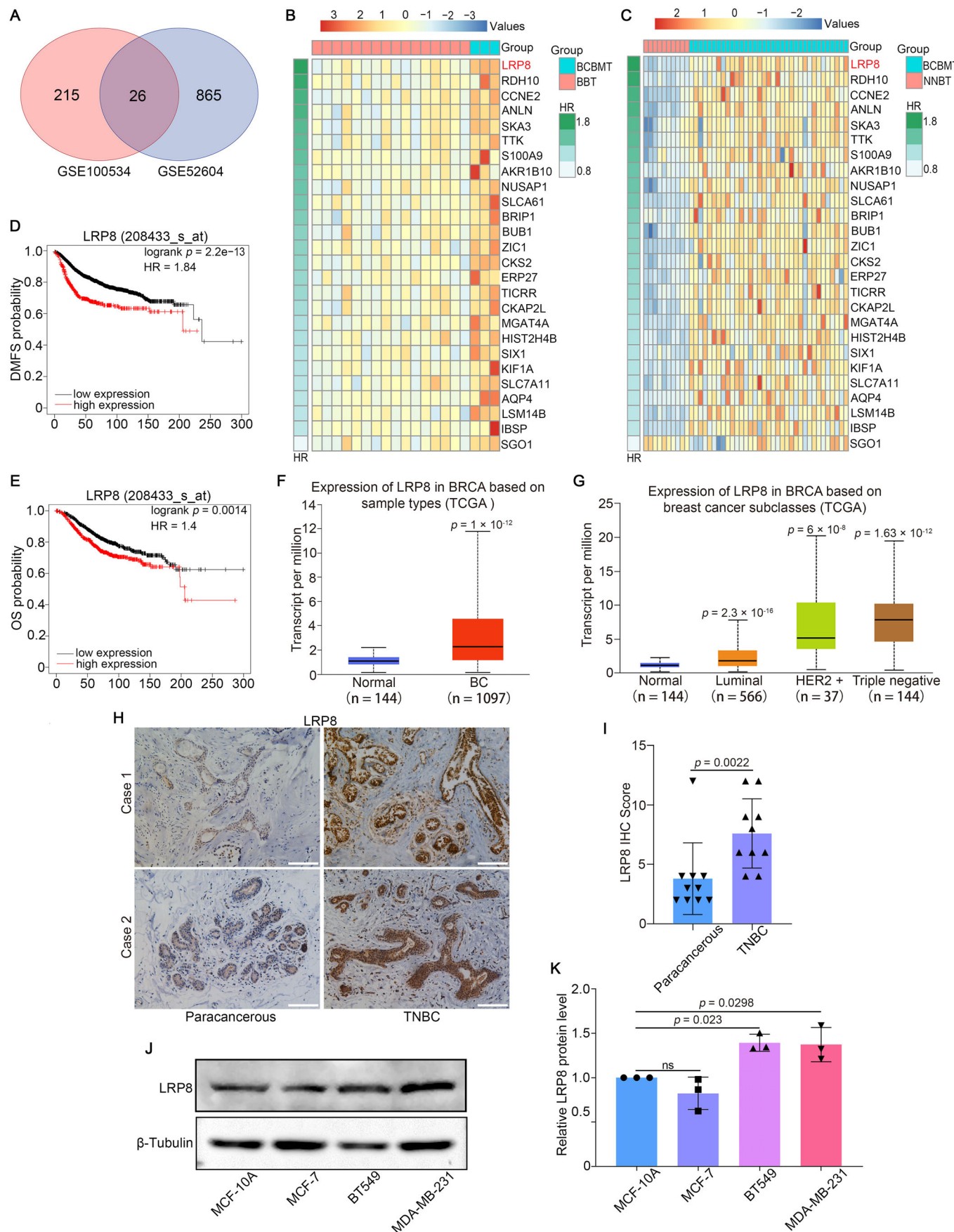

◄ **Figure 2. LRP8 is highly expressed in TNBC and predicts a high risk of brain metastasis.**

(A) A Venn diagram illustrates the co-expressed genes identified in two datasets (GSE100534 and GSE52604, $P < 0.01$, logFC $\geq 1.5$). (B, C) Heatmaps visualize 26 co-expressed genes from datasets GSE100534 (BTT vs. BCBMT) and GSE52604 (NNBT vs. BCBMT), organized according to the HR of DMFS derived from Kaplan–Meier analyses. (D, E) Kaplan–Meier survival curves depict DMFS and OS probabilities of BC patients, stratified by high and low expression levels of LRP8. (F) Box plots illustrate the expression levels of LRP8 across breast cancer patients and normal individuals. Normal (minimum: 0.127, Q1: 0.858, median: 1.085, Q3: 1.354, maximum: 2.207), BC (minimum: 0.126, Q1: 1.209, median: 2.267, Q3: 4.518, maximum: 11.792). $P = 1 \times 10^{-12}$. (G) Box plots illustrate the expression levels of LRP8 across various breast cancer subtypes and normal individuals. Normal (minimum: 0.127, Q1: 0.858, median: 1.085, Q3: 1.354, maximum: 2.207), luminal (minimum: 0.126, Q1: 1.045, median: 1.765, Q3: 3.269, maximum: 7.787), HER2 + (minimum: 0.497, Q1: 3.551, median: 5.129, Q3: 10.331, maximum: 20.231), triple negative (minimum: 0.404, Q1: 4.661, median: 7.863, Q3: 10.133, maximum: 19.513). $P = 2.3 \times 10^{-16}$ (normal vs. Luminal), $P = 6 \times 10^{-8}$ (normal vs HER2 + ), $P = 1.63 \times 10^{-12}$ (normal vs. triple negative). (H) Immunohistochemistry staining for LRP8 was performed on human TNBC tissues and paracancerous tissues. Scale bar: 100 μm. (I) IHC scores of (H) were presented. $n = 10$. (J) Western blot analysis was performed on MCF-10A, MCF-7, BT549, and MDA-MB-231 cell lines to evaluate LRP8 protein expression levels. (K) Quantitative analysis revealed the relative LRP8 protein level normalized to β-tubulin. $n = 3$. BTT breast tumor tissue, NNBT non-neoplastic breast tissue, BCBMT breast cancer brain metastasis tissue, DMFS distant metastasis-free survival, OS overall survival, HR hazard ratio. Data information: data are shown as mean ± SD. $P$ values were analyzed with unpaired Student's $t$ test (F), Mann–Whitney test (I) and one-way ANOVA test (G, K). ns non-significant. Source data are available online for this figure.

analysis on control (BM0) and BM6 cells was performed. Gene set enrichment analysis revealed that genes involved in cell migration and cytoskeleton organization were significantly upregulated in BM6 cells (Fig. 5J,K). Key genes such as VAV3 and CDC42BPG, which are downstream effectors of the Reelin-LRP8 signaling pathway, were also upregulated in BM6 cells (Fig. 5L). To experimentally validate the enhanced migratory and invasive capacities of BM6 cells, we performed transwell and wound-healing assays. These assays demonstrated that BM6 cells exhibited significantly increased migratory and invasive abilities compared to BM0 cells (Figs. 5M–O and EV4I,J). To further assess the in vivo metastatic potential of BM6 cells, we transplanted them into zebrafish. BM6 cells displayed significantly accelerated brain metastasis, as evidenced by increased migration distance and shorter time to brain colonization compared to BM0 cells (Fig. 5P,Q). These results suggest that the sixth-generation of brain metastatic cells exhibit enhanced migratory and invasive capabilities, likely due to upregulation of downstream effectors of the Reelin-LRP8 signaling pathway.

## MEN 10207 inhibits the Reelin-LRP8 signaling pathway

Given the role of the Reelin-LRP8 signaling pathway in regulating BCBM, we conducted molecular docking studies to identify potential LRP8 inhibitors. The screening revealed that onjisaponin B (OB) and MEN 10207 interact with the N-terminal extracellular ligand-binding domain of LRP8 (Table EV2). OB, a primary active compound derived from the traditional Chinese medicinal herb *Polygala*, is known for its neuroprotective effects (Peng et al, 2020), while MEN 10207 is a neurokinin-2 receptor antagonist (Luo and Wiesenfeld-Hallin, 1993). Detailed binding analyses demonstrated that OB forms hydrogen bonds with residues Cys186, Gly217, Ala220, Arg225, Glu235, and Asp236 of LRP8, whereas MEN 10207 interacts with Asp187, Asp189, Pro204, Ser624, Thr625, and Asp626 of LRP8, indicating stable ligand-receptor interactions (Fig. 6A). Molecular dynamics simulations further supported the stability of these interactions, with low root mean square deviation (RMSD) values (1.5-2.5 nm, Fig. 6B) and stable total potential energy profiles for the ligand-LRP8 complexes (Fig. 6C).

We next evaluated the effects of OB and MEN 10207 on the viability of MDA-MB-231 cells using CCK-8 assays. The IC50 values for OB and MEN 10207 were determined to be 10.96 μM and 19.37 μM at 24 h, respectively (Fig. EV5A,B). Based on these results,

concentrations of 5 μM OB and 10 μM MEN 10207 were selected for subsequent experiments. Wound-healing and transwell assays revealed that both compounds significantly inhibited the migration and invasion of MDA-MB-231 cells in vitro (Fig. EV5C–G).

To assess the safety of OB and MEN 10207 in a developing zebrafish model, these compounds were administered starting at 2 day post fertilization (dpf) and incubated at 36 °C. Both compounds were well-tolerated at lower concentrations, indicating favorable safety profiles (Fig. EV5H). In zebrafish xenograft models, treatment with OB or MEN 10207 significantly reduced GFP fluorescence intensity, indicating regression of cancer cell growth. Additionally, both compounds effectively inhibited the abluminal migration of cancer cells along the zebrafish vasculature (Fig. 6D–F).

We also evaluated the safety and toxicity profiles of OB and MEN 10207 (administered via intravenous injection) in ~8-week-old female nude mice. The results indicated that OB caused complete mortality (100%) within three days, suggesting it may not be suitable for use in mice (Fig. EV5I). In contrast, MEN 10207 had no significant impact on mortality rates, body weight, or serum levels of aspartate transaminase (AST), alanine transaminase (ALT), and creatinine (Cr), demonstrating a favorable safety profile in mice (Fig. EV5I,J; Table EV3). To further assess the therapeutic efficacy of MEN 10207 in inhibiting BCBM in mammals, we established a nude mouse xenograft model of BCBM and administered MEN 10207 to xenografted mice (Fig. 6G). Notably, the MEN 10207-treated group exhibited significantly smaller tumor sizes in the brain compared to the vehicle-treated group, underscoring its anti-BCBM efficacy in mammals (Fig. 6H,I; Movies EV5–7). To further investigate localization of brain metastatic MDA-MB-231 cells with blood vessels, we confirmed immunofluorescence analysis of brain sections of vehicle-treated mice stained with anti-CD31 antibody. We found the MDA-MB-231 cells were predominantly disseminated in the external of blood vessels of brain parenchyma (Fig. 6J). Next, we assessed the effects of MEN 10207 treatment in activation of CDC42. Western blot assay showed that MEN 10207 treatment decreased the level of GTP-bound CDC42 in MDA-MB-231 cells and inhibited the activation of CDC42 following Reelin treatment (Fig. 6K,L). In summary, our findings demonstrate that MEN 10207 effectively inhibits TNBC cell migration and invasion by targeting the Reelin-LRP8 signaling axis. These results highlight its potential as a promising therapeutic candidate for managing BCBM.

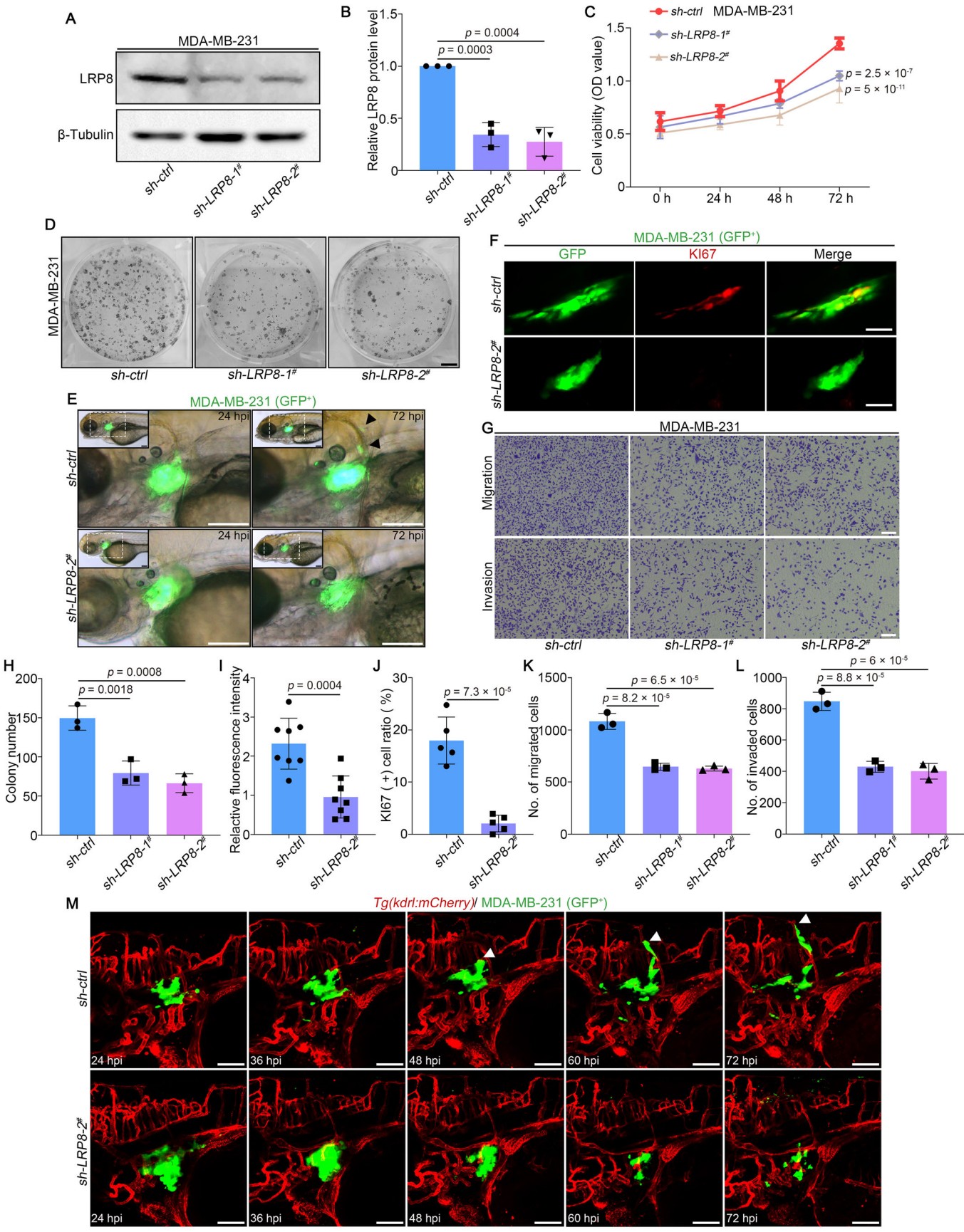

**Figure 3. LRP8 knockdown inhibits proliferation, migration and invasion of MDA-MB-231 cells.**

(A) MDA-MB-231 cells transfected with sh-ctrl, sh-LRP8-1# and sh-LRP8-2# were subjected to western blot assay to analyze the protein level of LRP8. (B) Quantitative analysis showed the relative LRP8 protein level normalized to β-tubulin, $n = 3$. (C) Cell viability of MDA-MB-231 cells was evaluated by the CCK-8 assays. $n = 3$. $P = 2.5 \times 10^{-7}$ (sh-ctrl vs. sh-LRP8-1#), $P = 5 \times 10^{-11}$ (sh-ctrl vs. sh-LRP8-2#). (D) Images of the number of colonies formed from MDA-MB-231 cells in three cell lines. Scale bar: 500 μm. (E) Images of GFP cancer cell proliferation in zebrafish after transplantation. Scale bar: 200 μm. (F) Following injection of MDA-MB-231 (GFP⁺) cells into zebrafish, whole-mount immunofluorescence staining for KI67 (red signal) and GFP was conducted on the embryos. Scale bar: 50 μm. (G) Transwell migration assay and Matrigel transwell invasion assay were performed in MDA-MB-231 cells transfected with sh-ctrl, sh-LRP8-1# and sh-LRP8-2#. Scale bar: 200 μm. (H) Quantification of the number of colonies formed from MDA-MB-231 cells in three cell lines. $n = 3$. (I) Quantitative analysis of E showed the relative fluorescence intensity of GFP in vivo. The measurement of fluorescence of GFP at 24 hpi was used as the baseline. $n = 8$. (J) Quantitative analysis of F showed the KI67 positive cell rate in two cell lines in vivo. $n = 5$. $P = 7.3 \times 10^{-5}$. (K) Quantitative analysis of G showed the migratory abilities in three cell lines of MDA-MB-231 cells. $n = 3$. $P = 8.2 \times 10^{-5}$ (sh-ctrl vs. sh-LRP8-1#), $P = 6.5 \times 10^{-5}$ (sh-ctrl vs. sh-LRP8-2#). (L) Quantitative analysis of G showed the invasive abilities in three cell lines of MDA-MB-231 cells. $n = 3$. $P = 8.8 \times 10^{-5}$ (sh-ctrl vs. sh-LRP8-1#), $P = 6 \times 10^{-5}$ (sh-ctrl vs. sh-LRP8-2#). (M) Real-time in vivo imaging was conducted at 24 hpi, 36 hpi, 48 hpi, 60 hpi and 72 hpi to monitor the dynamic behavior of MDA-MB-231 cells transfected with sh-ctrl and sh-LRP8-2#. The white arrow heads indicated the disseminated cancer cells. $n = 3$. Scale bar: 100 μm. Data information: data are shown as mean ± SD, P values were analyzed with one-way ANOVA test (B, H, K, L), unpaired Student's t test (I, J) and two-way ANOVA test (C). Source data are available online for this figure.

## Discussion

Tumor cells primarily utilize intravascular dissemination for metastatic spread. However, melanoma cells exhibit the ability to migrate to adjacent or distant sites without entering the vascular system, a process termed extravascular migratory metastasis (Barnhill et al, 2016; Lugassy et al, 2020, 2014). Notably, Sipkins et al, observed that acute lymphoblastic leukemia cells residing in vertebral or calvarial bone marrow can invade the leptomeninges by migrating along the laminin-rich external surfaces of emissary vessels (Yao et al, 2018). Recently, Sipkins identified, for the first time, that BC cells can migrate to the leptomeninges through abluminal migration via emissary veins (Whiteley et al, 2024), uncovering a novel route for BCBM. Given that extravascular migratory metastasis is a significant yet often overlooked phenomenon, we aim to establish an effective in vivo animal model to track the dynamic abluminal migration of brain metastatic BC cells.

Immunodeficient murine xenograft models serve as indispensable tools in breast cancer metastasis research, particularly for investigating brain metastatic mechanisms (Fernando et al, 2022). Current methodological advances demonstrate that BCBM models can be established through multiple approaches, for example: (1) Gan et al, successfully developed a brain metastasis model by intracardiac injection of brain-tropic metastatic TNBC cells into female mice (Gan et al, 2024); (2) Whiteley et al revealed through EO771-tdT cell engraftment in C57BL/6 mice that breast cancer cells circumvent the blood-brain barrier via migration along the abluminal surface of emissary veins to reach the leptomeningeal space (Whiteley et al, 2024); (3) Clinical translation models using patient-derived TNBC cells have been established through intracisternal or intracarotid arterial administration, while BT474 cell injection via the intracarotid artery provides another validated modeling strategy (Cordero et al, 2022; Kitamura et al, 2021). However, traditional xenograft models require extended durations, substantial financial investments, and pose significant challenges for monitoring dynamic in vivo progression (Gamble et al, 2021; Stoletov et al, 2007). In contrast, zebrafish possess a brief maturation period and rapid organogenesis (White et al, 2013), and they lack a fully developed adaptive immune system until ~28 dpf. This characteristic allows for efficient xenotransplantation of human cancer cells in zebrafish embryos without immune rejection (Novoa and Figueras, 2012). Moreover, the incorporation of fluorescently labeled tumor cells enables real-time observation of their proliferation, migration, and invasion within the zebrafish body (Berghmans et al, 2005). Consequently, our zebrafish xenograft model represents a valuable tool for investigating the dynamic abluminal migration of brain metastatic BC cells and the mechanisms driving extravascular migratory metastasis.

BC is classified into four clinical subtypes based on the expression of human epidermal growth factor 2 (HER2), hormone receptors (HR), and progesterone receptor (PR): Luminal A, Luminal B, HER2-positive, and TNBC (Hung et al, 2023). Among these, TNBC is associated with the worst prognosis and exhibits the highest aggressiveness, with up to 46% of TNBC patients developing BM (Lin et al, 2012). Notably, TNBC shows a strong propensity for leptomeningeal metastasis (Franzoi and Hortobagyi, 2019). Laminin is abundantly present in endothelial cells during zebrafish development (Eve and Smith, 2017), and integrin α6 is the second most abundant mRNA in the MDA-MB-231 cell line relative to other human BC cell lines, according to The Human Protein Atlas. Given that BC cells expressing high levels of integrin α6 can adhere to laminin-rich surfaces, we transplanted MDA-MB-231 (GFP⁺) cells into Tg(kdrl:mCherry) zebrafish at 48 h post-fertilization (hpf). Our findings revealed that MDA-MB-231 cells could adhere to the posterior cardinal veins (PCeVs) and migrate along their external surfaces, ultimately reaching the zebrafish brain. Furthermore, RNA sequencing indicated that integrin α6 expression was significantly elevated in BM6 cells compared to control (BM0) cells. We hypothesize that the high expression of integrin α6 facilitates the adhesion of MDA-MB-231 cells to PCeVs, prompting further investigation into the mechanisms underlying abluminal migration of TNBC cells.

LRP8 has been implicated in various malignancies, with studies showing that its overexpression increases phospho-STAT3 (p-STAT3) levels, a marker associated with metastasis and poorer outcomes in osteosarcoma (Zheng et al, 2021). Additionally, miR-30b-5p has been shown to reduce the viability, migration, and invasion of lung cancer cells by targeting LRP8, suggesting that miR-30b-5p may inhibit lung cancer progression through LRP8 modulation (Qiu et al, 2021). In our study, we identified LRP8 as a co-expressed gene linked to poorer survival outcomes in BC patients through analysis of the GSE100534 and GSE52604 microarray datasets. Western blotting and immunohistochemistry demonstrated that LRP8 is significantly overexpressed in TNBC cell lines compared to MCF-7 and MCF-10A, as well as in TNBC

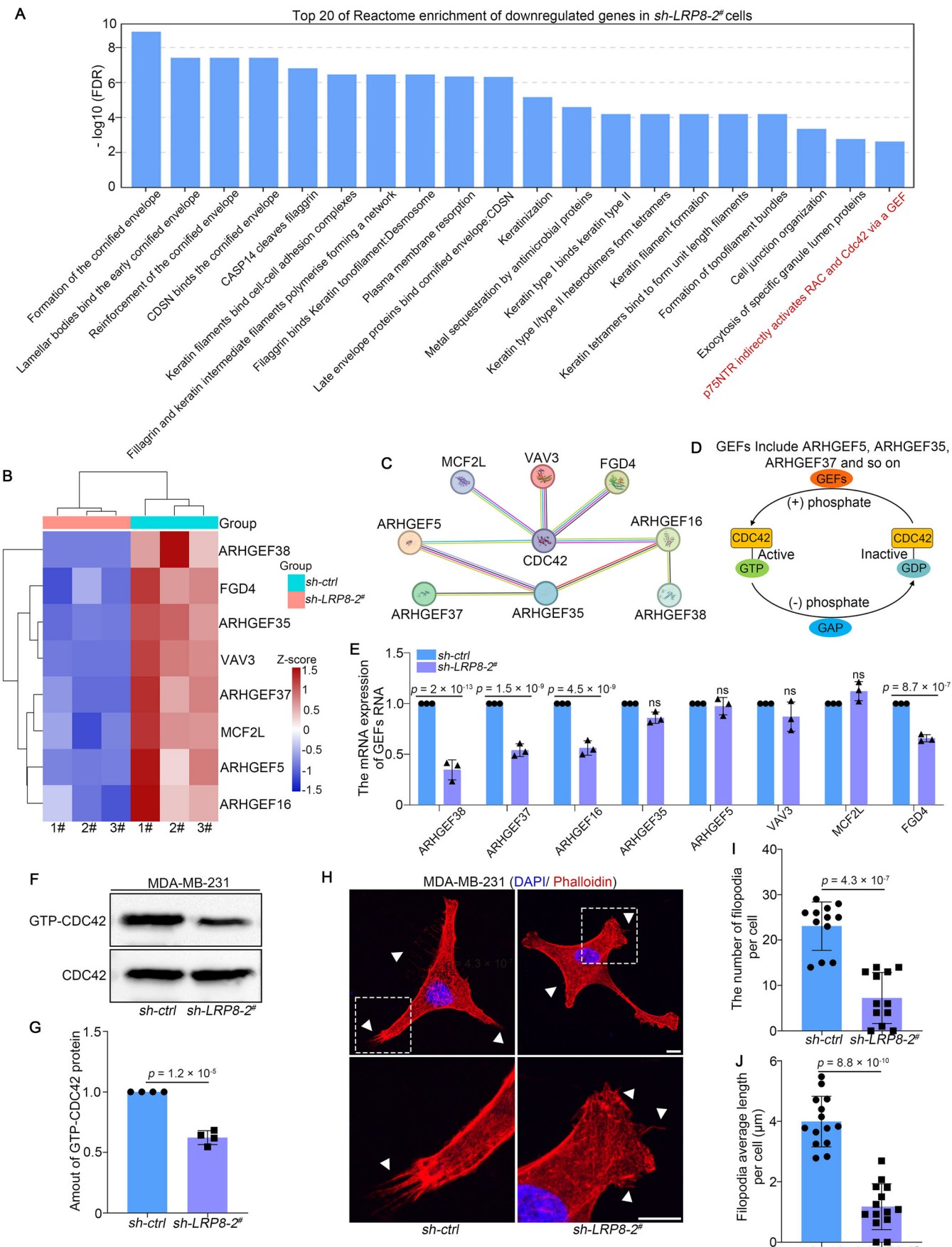

**A** Top 20 of Reactome enrichment of downregulated genes in *sh-LRP8-2*# cells

**B**

**C**

**D** GEFs Include ARHGEF5, ARHGEF35, ARHGEF37 and so on

**E**

**F** MDA-MB-231

**G**

**H** MDA-MB-231 (DAPI/ Phalloidin)

**I**

**J**

**Figure 4. LRP8 knockdown decreases the GTP-CDC42 expression to inhibit filopodia formation.**

(A) Top 20 of Reactome enrichment of downregulated genes in MDA-MB-231 cells transfected with *sh-ctrl* and *sh-LRP8-2*#. FC > | 2 | and FDR < 0.05. (B) The heatmap visualization analyses demonstrating the significantly differentially expressed GEFs in MDA-MB-231 cells transfected with *sh-ctrl* and *sh-LRP8-2*#. (C) The PPI network analysis conducted by STRING revealed the interaction of CDC42 and GEFs. (D) The classical Rho GTPases cycle involves both an inactive GDP-bound form and an active GTP-bound form. (E) Quantitative real-time PCR (qRT-PCR) assays were used to measure the mRNA levels of GEFs in MDA-MB-231 cells transfected with *sh-ctrl* and *sh-LRP8-2*#. $n = 3$. $P = 2 \times 10^{-13}$ (ARHGEF38), $P = 1.5 \times 10^{-9}$ (ARHGEF37), $P = 4.5 \times 10^{-9}$ (ARHGEF16), $P = 8.7 \times 10^{-7}$ (FGD4). (F, G) MDA-MB-231 cells transfected with *sh-ctrl* and *sh-LRP8-2*# were subjected to western blot assay to analyze the protein level of activated GTP-bound CDC42. $n = 4$. $P = 1.2 \times 10^{-5}$. (H) MDA-MB-231 cells transfected with *sh-ctrl* and *sh-LRP8-2*# were stained with TRITC-labelled phalloidin (red) and DAPI (blue) to detect the filopodia of MDA-MB-231 cells. The white arrow heads indicated the filopodia of MDA-MB-231 cells. Scale bar: 10 μm. (I, J) Quantification analysis of the number of filopodia per cell ($n = 12$, $P = 4.3 \times 10^{-7}$) and average filopodia length ($n = 14$, $P = 8.8 \times 10^{-10}$) of MDA-MB-231 cells transfected with *sh-ctrl* and *sh-LRP8-2*#. FC: fold change. FDR: false discovery rate. Data information: data are shown as mean ± SD, P values were analyzed with unpaired Student's *t* test (G, I, J) and two-way ANOVA test (E). ns non-significant. Source data are available online for this figure.

primary tissues relative to adjacent non-cancerous tissues. Given that TNBC cells exhibit heightened migratory and invasive capabilities, LRP8 may play a critical role in the brain metastasis of TNBC patients. Therefore, understanding the relationship between LRP8 and BCBM is of utmost importance, particularly since the role of LRP8 in regulating BCBM remains unclear. Our experiments involving *LRP8* knockdown in MDA-MB-231 and BT549 cells demonstrated that *LRP8* depletion inhibited cellular proliferation, migration, and invasion in vitro, as well as abluminal migration toward the zebrafish brain in vivo. Moreover, *LRP8* knockdown reduced filopodia formation in MDA-MB-231 cells. These findings position LRP8 as a high-risk gene associated with BCBM, highlighting its critical role in the abluminal migration of TNBC cells.

Recent studies suggest that the brain microenvironment releases various factors that may promote the growth and movement of BC cells within the brain (Padmanaban et al, 2024; Whiteley et al, 2024). Reelin, which is predominantly expressed in GABAergic interneurons and glutamatergic granule cells, plays a crucial role in neuromorphogenesis by binding to its receptor LRP8 (Impagnatiello et al, 1998; Reddy et al, 2011). The Reelin signaling pathway regulates actin dynamics via LIMK1-mediated inhibition of cofilin, influencing cytoskeletal dynamics and cellular motility (Santana and Marzolo, 2017). Additionally, Reelin binding to LRP8 modulates phosphatidylinositol-3-kinase (PI3K) signaling, activating CDC42 and RAC1 to enhance filopodia and branch formation during neurodevelopment (Leemhuis and Bock, 2011). Notably, Reelin secreted by small cell lung cancer (SCLC) cells has been shown to recruit astrocytes to brain metastases, promoting SCLC proliferation (Qu et al, 2023). Thus, high levels of Reelin in the brain may serve as a catalyst for cancer progression. To further investigate the relationship between Reelin and LRP8 in BCBM, we administered recombinant Reelin protein to assess its effects on MDA-MB-231 cells. Our in vitro assays revealed that recombinant Reelin significantly promoted the migration and invasion of MDA-MB-231 cells, while *LRP8* knockdown cells exhibited no such response. In vivo experiments indicated a reduction in the brain metastatic potential of MDA-MB-231 cells following treatment with an anti-Reelin antibody compared to an isotype control. Given that CDC42-GTP promotes F-actin polymerization and filopodia formation (Jacquemet et al, 2016), facilitating cancer cell migration and survival at distant metastatic sites (Shibue et al, 2012), we found that Reelin treatment upregulated GTP-CDC42 in control cells, with no change observed in *LRP8* knockdown cells. These

results suggest that Reelin enhances the migratory and invasive capabilities of MDA-MB-231 cells through LRP8, modulating CDC42 activation to promote filopodia formation during BC progression.

We also successfully isolated brain-colonizing metastatic cancer cell populations, analyzed mRNA expression levels of LRP8 in BM6 and BM0 cells, and conducted RNA sequencing. Not only high levels of LRP8 expression in TNBC tissues and cell lines, but we also observed elevated LRP8 levels in BM6 cells compared to BM0 cells. Additionally, biological process analysis indicated enhanced cell migration and cytoskeletal organization in BM6 cells, with significant upregulation of key genes such as VAV3 and CDC42BPG, which are crucial for GTPase cycling of CDC42 and cytoskeletal reorganization, respectively. This suggests that the Reelin-LRP8-CDC42 signaling pathway may be augmented in brain metastatic cell populations. Moreover, we identified MEN10207 as a potential inhibitor of LRP8 protein that effectively suppressed migration of MDA-MB-231 cells in vitro and in vivo. As MEN 10207 interacted with the extracellular Reelin-binding domain of LRP8 protein (from Asp46 to Leu334) (Yasui et al, 2010), this compound may target Reelin-LRP8-CDC42 signaling axis to inhibit migration of TNBC cells.

In conclusion, our study established a zebrafish xenograft model and demonstrated that MDA-MB-231 cells migrated abluminally along the external surfaces of PCeVs. Utilizing this model, we explored the functional roles and underlying mechanisms of genes associated with BCBM. Our findings highlight the critical role of LRP8 in regulating TNBC cell proliferation, migration, and invasion in vitro, as well as its facilitation of abluminal migration toward the brain in vivo. Furthermore, we identified the Reelin-LRP8 signaling pathway as a key regulatory mechanism in aggressive brain metastatic breast cancer, with LRP8 enhancing CDC42 activation to promote filopodia formation, dependent on Reelin. This zebrafish model provides a valuable platform for investigating the mechanisms of extravascular migratory metastasis, while the Reelin-LRP8-CDC42 signaling axis may contribute significantly to the abluminal migration of brain metastatic BC cells. What's more, therapeutic interventions using MEN 10207 effectively inhibited MDA-MB-231 cell migration and activation of CDC42 in vitro and inhibited BM of MDA-MB-231 cells in both zebrafish and mouse xenograft models, suggesting MEN 10207 could be considered as a novel potential candidate for BCBM therapy by targeting Reelin-LRP8-CDC42 signaling axis.

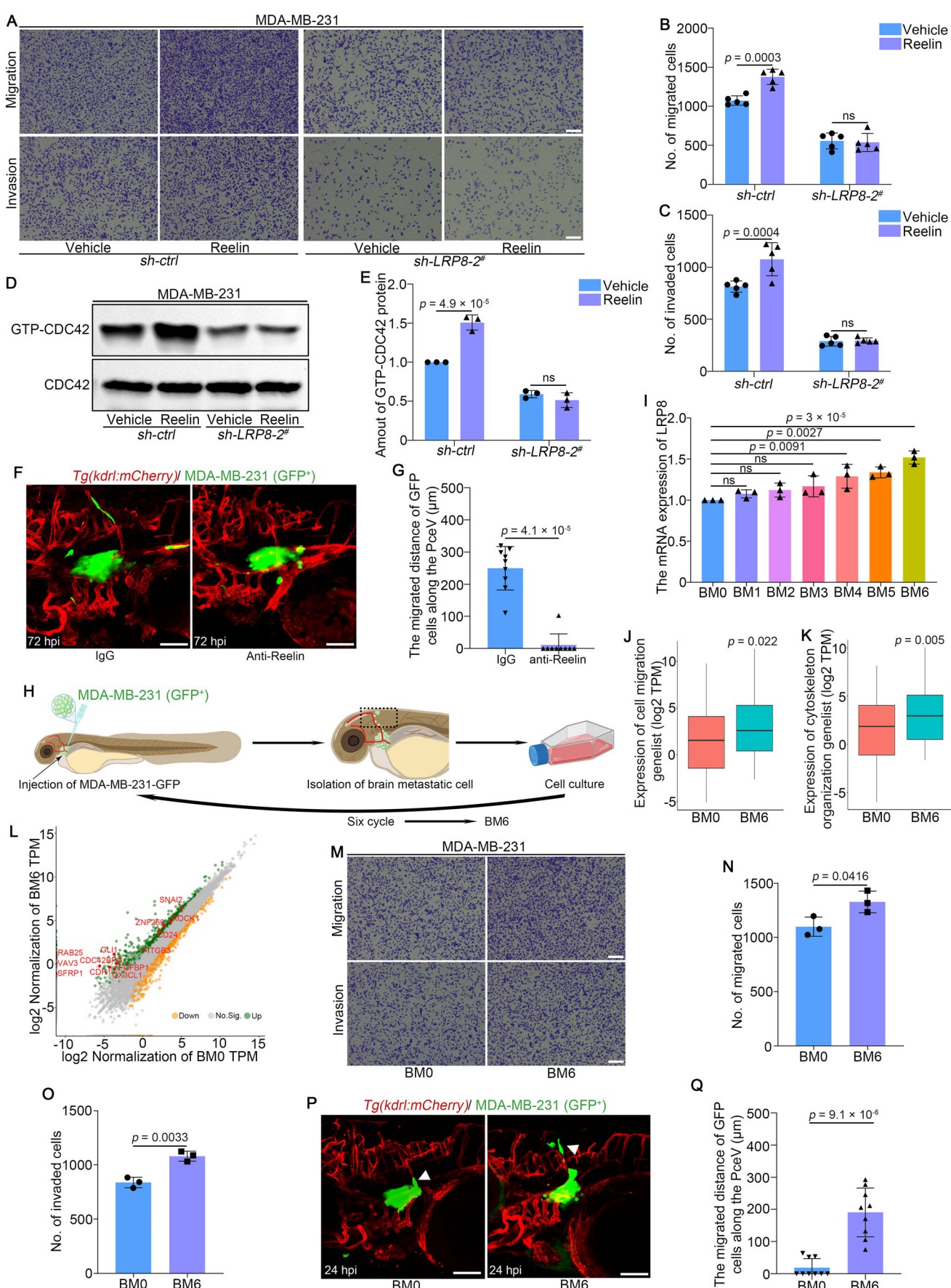

◄ **Figure 5. Reelin-LRP8 signaling pathway may activate CDC42 to regulate migration and invasion of MDA-MB-231 cells.**

(A) Transwell migration assay and Matrigel transwell invasion assay were performed in MDA-MB-231 cells transfected with *sh-ctrl* and *sh-LRP8-2#* following Reelin treatment. Scale bar: 200 µm. (B, C) Quantity of migrated and invasive cells. $n = 5$. (D) MDA-MB-231 cells transfected with *sh-ctrl* and *sh-LRP8-2#* were subjected to western blot assay to analyze the protein level of activated GTP-bound CDC42, following Reelin treatment. (E) Quantitative analysis revealed the relative GTP-bound CDC42 level normalized to total CDC42. $n = 3$. $P = 4.9 \times 10^{-5}$ (*sh-ctrl*). (F, G) Images and quantification analysis of the migrated distance of brain metastatic cells along PceV in zebrafish models at 72 hpi after the anti-Reelin monoclonal antibody treatment. $n = 9$. Scale bar: 100 µm. (H) The schematic representation illustrates the isolation of brain metastatic cells after the transplantation of MDA-MB-231 (GFP+) cells into zebrafish. (I) The mRNA expression of LRP8 in various brain metastatic cell populations. $n = 3$. $P = 0.0091$ (BM0 vs. BM4), $P = 0.0027$ (BM0 vs. BM5), $P = 3 \times 10^{-5}$ (BM0 vs. BM6). (J) The expression of cell migration related genes from GO analysis. $n = 86$. BM0 (minimum: 0, Q1: 0.097, median: 1.377, Q3: 14.297, maximum: 871.831), BM6 (minimum: 0.161, Q1: 1.319, median: 5.963, Q3: 38.809, maximum: 2572.556). FC > | 2 | and FDR < 0.05. (K) The expression of cytoskeleton organization related genes from GO analysis. $n = 87$. BM0 (minimum: 0, Q1: 0.2095, median: 2.637, Q3: 14.2505, maximum: 282.875), BM6 (minimum: 0.327, Q1: 1.415, median: 7.716, Q3: 34.227, maximum: 1053.12). FC > | 2 | and FDR < 0.05. (L) Scatter plot of significant differentially expressed genes between BM0 vs. BM6. (M) Transwell migration assay and Matrigel transwell invasion assay were performed in BM0 cells and BM6 cells, Scale bar: 200 µm. (N, O) Quantification analysis of migrated and invaded BM0 or BM6 cells. $n = 3$. (P) Images of BM0 and BM6 cells migrating along PceV in zebrafish at 24 hpi. The white arrow heads indicated the disseminated cancer cells. Scale bar: 100 µm. (Q) Quantification analysis of migrated distance of brain metastatic cells along PceV in zebrafish. $n = 9$. $P = 9.1 \times 10^{-6}$. FC fold change, FDR false discovery rate. Data information: data are shown as mean ± SD, P values were analyzed with paired Student's *t* test (J, K), unpaired Student's *t* test (N, O, Q), two-way ANOVA test (B, C, E), Mann–Whitney test (G) and one-way ANOVA test (I). ns non-significant. Source data are available online for this figure.

# Methods

## Reagents and tools table

| Reagent/resource | Reference or source | Identifier or catalog number |
|---|---|---|
| **Experimental models** | | |
| BALB/c Nude (*M. musculus*) | Beijing Vital River Laboratory Animal Technology Co., Ltd.(Beijing, China) | NA |
| *Tg(kdrl:mCherry)* zebrafish | National Aquatic Biological Resource Center, China | NA |
| **Recombinant DNA** | | |
| *pLL3.7 vector* | Affiliated Hospital of Guangdong Medical University, China | NA |
| **Antibodies** | | |
| Rabbit anti-LRP8 antibody | Boster | #A03444-2 |
| Mouse anti-β-tubulin antibody | ZSGB-BIO | #TA-10 |
| Goat anti-rabbit HRP | ZSGB-BIO | #ZB-5301 |
| Goat anti-mouse HRP | ZSGB-BIO | #ZB-2305 |
| Mouse anti-Cdc42-GTP antibody | New East Biosciences | #26905 |
| Mouse anti-Cdc42 antibody | Proteintech Group | #67212 |
| Mouse anti-Reelin monoclonal antibody | Abcam | #ab78540 |
| Mouse IgG antibody | ABclonal | #AC011 |
| Rabbit Anti-KI67 antibody | Proteintech Group | #27309-1-AP |
| Alexa Fluor® 594 AffiniPure® Goat Anti-Rabbit IgG | Jackson | #111-585-046 |
| Alexa Fluor® 647 AffiniPure® Goat Anti-Rabbit IgG | Jackson | #111-605-144 |
| DAPI | Sigma | #28718-90-3 |
| TRITC-labelled phalloidin | Thermo Fisher Scientific | #R415 |

| Reagent/resource | Reference or source | Identifier or catalog number |
|---|---|---|
| Rabbit anti-CD31 monoclonal antibody | Abcam | #ab222783 |
| **Oligonucleotides and other sequence-based reagents** | | |
| qPCR primers | This study | See Table EV4 |
| Targeting sequences of *sh-LRP8* and *sh-ctrl* | This study | See "Methods" |
| **Chemicals, enzymes and other reagents** | | |
| CCK-8 Kit | DOJINDO | #CK04 |
| Paraformaldehyde | Sigma | #p6148 |
| DMEM/F 12 medium | Cell Cook | #CC4007S |
| RPMI 1640 medium | YEASEN | #41402ES76 |
| Insulin | Beyotime | #P3376-100IU |
| Fetal bovine serum | Lonsera | #S712-012S |
| High glucose DMEM medium | YEASEN | #41401ES76 |
| Trypsin-EDTA | Gbico | #25200-056 |
| Crystal violet | Aladdin | #C110703 |
| Matrigel matrix | BD | #3356234 |
| Protein A/G agarose | New East Biosciences | #30301 |
| EasyBlot kit | GeneTex | #GTX425858 |
| Recombinant human Reelin | RD systems | #8456-MR-050 |
| Onjisaponin B | MCE | #HY-N2099 |
| MEN 10207 | MCE | #HY-151413 |
| PTU | Sigma | #P7629 |
| Lipofectamine 3000 | Thermo Fisher Scientific | #L3000015 |
| Polyethylenimine | FUSHENBIO | #FSF0002 |
| Polybrene | Solarbio | #H8761 |
| IHC Kit | Sangon Biotech | #D601037-0050 |
| Tricaine | Sigma | #E10521 |

| Reagent/resource | Reference or source | Identifier or catalog number |
|---|---|---|
| Agarose | Biosharp | #BS081 |
| Antibiotic-antimycotic solution | Beyotime | #C0224 |
| TRIZOL reagent | Invitrogen | #15596018CN |
| SYBR Green Master | YEASEN | #11202ES08 |
| SDS-PAGE | Beyotime | #P0012A |
| ECL detection kit | Thermo Fisher Scientific | #34577 |
| Protein Marker (10-180 kDa) | GenStar | #M221 |
| Low melting agarose | Thermo Fisher Scientific | #16520100 |
| DMSO | Solarbio | #D8371 |
| Methylene blue | Aladdin | #M134389 |
| Alexa Fluor™ 647 cadaverine | Thermo Fisher Scientific | #A30679 |
| Tribromoethanol | Nanjing Aibei Biotechnology | #M2960 |
| PEG300 | MCE | #HY-Y0873N |
| Quadrol | Sigma | #122262 |
| Tert-Butanol | Sigma | #360538 |
| Benzyl benzoate | Sigma | #W213802 |
| Bisphenol-A ethoxylate diacrylate Mn 468 | Sigma | #413550 |
| Tween-80 | MCE | #HY-Y1891 |
| OCT | SAKURA | #4853 |
| Hifair® III 1st Strand cDNA Synthesis Kit (gDNA digester plus) | YEASEN | #11119ES60 |
| Triton X-100 | YuBioLab | #A600198 |
| PBS | SveviceBio | #G0002 |
| **Software** | | |
| Imaris 10.1 | Bitplane | NA |
| prism9.5 | GraphPad | NA |
| Fiji | Image J | NA |
| Adobe Photoshop 2022 | Adobe Systems | NA |
| Adobe Illustrator 2020 | Adobe Systems | NA |
| SnapGene 7.1 | Insightful Science | NA |
| **Other** | | |
| Light-sheet fluorescence microscopy | Nuohai Life Science | NA |
| Confocal microscope | Olympus | NA |
| Confocal microscope | Nikon | NA |
| Fluorescence stereomicroscope | Leica | NA |
| Cell Imaging Multifunctional Microplate Detection System | BioTek Cytation | NA |
| Microplate reader | BioTek Epoch | NA |
| Analytikjena QTower3G | Analytikjena | NA |

| Reagent/resource | Reference or source | Identifier or catalog number |
|---|---|---|
| CellTram Vario oil-pressure microinjector | Eppendorf | NA |
| Manual rotary microtome | Leica | NA |
| Fluorescence stereomicroscope | M-shot | NA |
| CM1950 cryostat | Leica | NA |

## Databases analysis from Gene Expression Omnibus, UALCAN databases and The Human Protein Atlas

We queried the Gene Expression Omnibus (GEO) database (https://www.ncbi.nlm.nih.gov/geo/) to access microarray data corresponding to accession numbers GSE100534 and GSE52604. We downloaded the LRP8 expression of BC patients with various subtypes compared with normal individuals from the UALCAN database (http://ualcan.path.uab.edu). Additionally, we accessed the integrin α6 expression of various BC cell lines from the Human Protein Atlas (https://www.proteinatlas.org/).

## Kaplan–Meier survival curve analysis

The prognostic significance of LRP8 in BC patients was evaluated using the Kaplan–Meier plots (http://kmplot.com/analysis/). The cases in the database were stratified based on LRP8 expression levels, with the bottom 75% classified as low expression and the top 25% as high expression. The HR and log rank $P$ value were computed. The HR greater than 1 indicated a poor prognosis, while the HR below 1 indicated a favorable prognosis.

## Cell culture

The human TNBC cell line labeled with GFP, MDA-MB-231 (GFP$^+$), was provided by Dr. Shouyu Wang's lab from the First Affiliated Hospital of Anhui Medical University, China. The human TNBC cell line BT549 was provided by Dr. Hua Zhang's lab from Guangdong Medical University, China. The human breast epithelial cell line MCF-10A and the human estrogen responsive breast cancer cell line MCF-7 were from Dr. Yuanqi Zhang's lab in the Affiliated Hospital of Guangdong Medical University, China. The MCF-10A cell line was grown in DMEM/F 12 medium supplemented with 20 ng/ml EGF, 5% horse serum, 0.5 μg/ml hydrocortisone and 10 μg/ml insulin (Cell Cook, CC4007S), BT549 cell line was grown in RPMI 1640 medium (YEASEN, 41402ES76) supplemented with 1.1 μg/ml insulin (Beyotime, P3376-100IU) and 10% fetal bovine serum (Lonsera, S712-012S) and the other cell lines were grown in high glucose DMEM medium (YEASEN, 41401ES76) supplemented with 10% fetal bovine serum. All cell lines incubated in 5% $CO_2$ incubator at 37 °C.

## Cell transfection and lentivirus transduction

Targeting sequences *sh-LRP8-1*# (5′-GCCTAATGGAGGCTGT-GAATA-3′), *sh-LRP8-2*# (5′-GACGAAGATGAGCTCCATATA-3′)

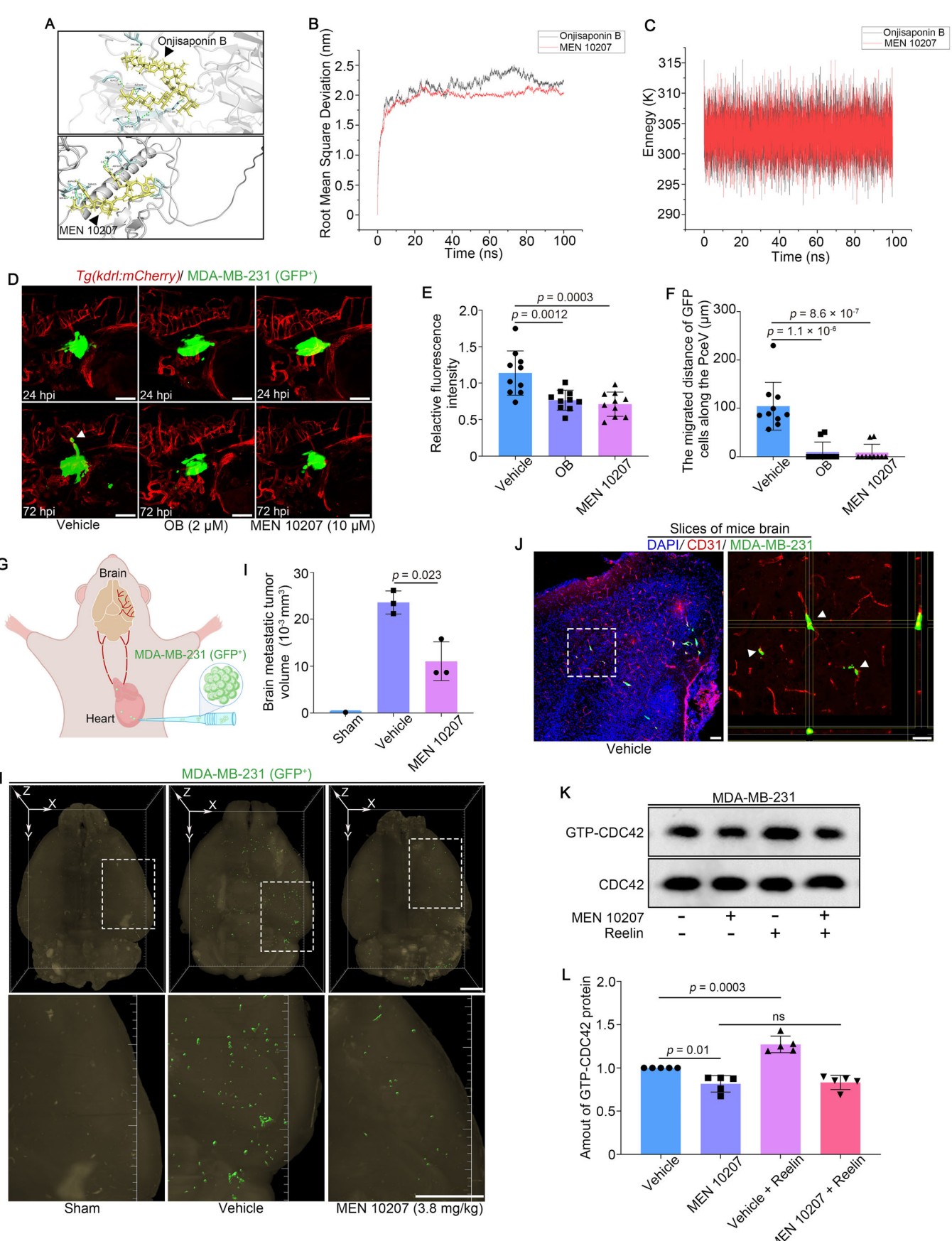

**Figure 6.  Onjisaponin B and MEN 10207 treatment inhibit the migration of MDA-MB-231 cells in vivo.**

(A) Three-dimensional binding pattern diagram of two candidate compounds and LRP8 protein. (B, C) The root mean square deviation fluctuations and potential energy fluctuations of compounds with LRP8 protein. (D) Representative confocal microscopy images of zebrafish xenograft models treated with vehicle, OB and MEN 10207. The white arrow heads indicated the disseminated cancer cells. Scale bar: 100 μm. (E) Quantitative analysis of D showed the relative fluorescence intensity of GFP in vivo. The measurement of fluorescence of GFP at 24 hpi was used as the baseline. $n = 10$. $P = 0.0012$ (Vehicle vs. OB), $P = 0.0003$ (Vehicle vs. MEN 10207). (F) Quantification analysis of the migrated distance of brain metastatic cells along PceV in zebrafish at 72 hpi was conducted. $n = 10$. $P = 1.1 \times 10^{-6}$ (Vehicle vs. OB), $P = 8.6 \times 10^{-7}$ (Vehicle vs. MEN 10207). (G) Schematic representation illustrates the nude mice xenograft model using MDA-MB-231 (GFP$^+$) cells. (H) Representative light-sheet fluorescence microscopy images of whole brains of nude mice xenograft models treated with vehicle, and MEN 10207. Scale bar: 1000 μm. (I) Quantitative analysis of H showed the brain metastatic tumor volume in vivo. $n = 3$. (J) Immunofluorescence staining (scale bar: 100 μm) for CD31 (red signal) and DAPI (blue signal) was conducted on mice brain section and three-dimensional imaging (scale bar: 50 μm) was performed to visualize the interaction between MDA-MB-231 cells and the blood vessels in mouse brain. MDA-MB-231 cells are displayed in green. The white arrow heads indicated the disseminated cancer cells in the mice brain. (K) MDA-MB-231 cells treated with vehicle and MEN 10207 were subjected to western blot assay to analyze the protein level of activated GTP-bound CDC42, following Reelin treatment. (L) Quantitative analysis revealed the relative GTP-bound CDC42 level normalized to total CDC42 protein. $n = 5$. Data information: data are shown as mean ± SD, P values were analyzed with unpaired Student's t test (I, vehicle-treated group vs. MEN 10207-treated group) and one-way ANOVA test (E, F, L). ns: non-significant. Source data are available online for this figure.

and *sh-ctrl* (5′-*CAACAAGATGAAGAGCACCAA*-3′) were inserted into *pLL3.7* vector. The transfection of MDA-MB-231 cells was performed using Lipofectamine 3000 (Thermo Fisher Scientific, L3000015) in accordance with the manufacturer's protocol.

For lentivirus transduction, these plasmids were transfected into HEK293T cells alongside the packaging vector *pGag-Pol*, *pRev* and *pVSVG* (Yubo Biotech, Shanghai, China) using polyethylenimine (PEI, FUSHENBIO, FSF0002). Culture medium containing lentivirus was harvested at 48 h post-transfection, filtered through a hydrophilic polyethersulfone membrane with 0.45 μm pores (Millipore, HAWP02500), and subsequently infected MDA-MB-231 cells with 4 μg/ml polybrene (Solarbio, H8761).

## Clinical sample collection and immunohistochemistry (IHC)

The ethical approval of experiments involved in clinical samples was approved by the Clinical Research Ethics Committee of the Affiliated Hospital of Guangdong Medical University (No.PJKT2024-169). Informed consent forms were signed by all participants prior to the study. Patients involved in this study were diagnosed with TNBC. All tissue samples were embedded in paraffin and sliced into 10 μm-thick sections using Leica manual rotary microtome (Nussloch, Germany). IHC assay was performed to evaluate the LRP8 expression using the IHC Kit of Sangon Biotech (D601037-0050) according to the manufacturer's protocol. The anti-LRP8 antibody (1:250, Boster, A03444-2) was used as the primary antibody. The expression of LRP8 was quantified by the immunoreactivity score (IRS), calculated by multiplying the proportion of positively stained cells by the intensity of the staining. All immunostained sections were scanned by an Olympus microscope (Tokyo, Japan). The percentage of positive stained cells was categorized on a scale from 0 to 4: 0 (<10%), 1 (10%–25%), 2 (26%–50%), 3 (51%–75%) and 4 (>75%). The staining intensity was assessed on a scale from 0 to 3:0 (negative), 1 (weak/light yellow), 2 (moderate/brown yellow) and 3 (strong/tan).

## Zebrafish husbandry

The adult zebrafish and embryos were maintained in accordance with standard laboratory procedures (Webb and Miller, 2006; Westerfield, 2000). The adult transgenic line *Tg(kdrl:mCherry)* zebrafish that express red fluorescent protein in the vascular

endothelial cells was used to visualize the vascular system (Cross et al, 2003). At 24 h post-fertilization (hpf), 0.003% (w/v) phenylthiocarbamide/N-phenylthiourea (PTU, Sigma-Aldrich, P7629) was added into the embryo water to inhibit the synthesis of pigments. All zebrafish in vivo experimental procedures were ethically compliant and approved by the Experimental Animal Ethics Committee of Guangdong Medical University. Adherence to the "Guangdong Laboratory Animal Management Regulations" was strictly observed during the handling of zebrafish.

## Establishment of zebrafish xenograft model

At 24 hpf, *Tg(kdrl:mCherry)* zebrafish embryos were anesthetized using a 0.2% solution of tricaine (Sigma, E10521) and then placed on a cell dish coated with 2% w/v agarose (Biosharp, BS081). A cell suspension containing 300-500 MDA-MB-231 (GFP$^+$) cells per embryo at a density of 3 cells/nl was microinjected into the perivitelline space of zebrafish using a microinjection needle (1.0 mm × 0.6 mm) and CellTram Vario oil-pressure microinjector (Eppendorf, Hamburg, Germany). The injected embryos were incubated in an incubator at 28.5 °C. Embryos were transferred to a 36 °C incubator after 6 h post-injection (hpi), for subsequent imaging at 24, 36, 48, 50, 72 hpi and so on.

## Isolation of brain metastatic cell populations

Following the transplantation of MDA-MB-231 (GFP$^+$) cells into zebrafish, tumor progression was assessed through daily fluorescence imaging. Brain metastatic lesions were subsequently examined, excised and minced using the M-shot fluorescence stereomicroscope (Guangzhou, China) under sterile conditions. Then the GFP-labeled tumor cells were isolated and transferred into DMEM supplemented with antibiotic-antimycotic solution (Beyotime, C0224) and 15% fetal bovine serum, denoting this cell line as first-generation of brain metastatic (BM1) cells. Following dissociation and subsequent expansion in culture, BM1 underwent additional five rounds of transplantation and isolation, denoting this cell line as BM6 cells.

## Quantitative real-time PCR (qRT-PCR)

Total RNA was isolated from the cells using TRIZOL reagent (Invitrogen, 15596018CN) and then reverse-transcribed into

complementary DNA (cDNA) with the provided Mix (YEASEN, 11119ES60). The qRT-PCR assay was performed to evaluate the mRNA expression according to the manufacturer's instructions (SYBR Green Master from YEASEN, 11202ES08). The qRT-PCR was conducted using the Analytikjena QTower3G (Jena, Germany). The complete list of primers used in this experiment can be found in Table EV4.

## Western blot analysis

Equal quantities of cellular proteins were separated using 8–12% sodium dodecyl sulfate polyacrylamide gel electrophoresis (SDS-PAGE, Beyotime, P0012A) and subsequently transferred onto polyvinylidene difluoride (PVDF, Millipore, ISEQ000) membranes. The PVDF membranes were then blocked with 5% skim milk for 2 h and incubated with primary antibodies overnight at 4 °C. Next, the membranes were incubated with HRP-conjugated secondary antibodies for 2 h at room temperature. Afterward, the resulting bands were visualized using an ECL detection kit (Thermo Fisher Scientific, 34577). The following antibodies were used: anti-LRP8 antibody (1:1000, Boster, A03444-2), anti-β-tubulin antibody (1:2000, ZSGB-BIO, TA-10), Goat anti-rabbit HRP (1:4000, ZSGB-BIO, ZB-5301) and goat anti-mouse HRP (1:4000, ZSGB-BIO, ZB-2305).

## Cell proliferation assay

Cell viability was assessed using the CCK-8 assay (DOJINDO, CK04) following the manufacturer's instructions. The absorbance was measured at 450 nm with a microplate reader (BioTek Epoch, USA), using wells devoid of cells as control blanks. Cell proliferation was quantified based on the absorbance values.

## Wound healing assay

Cells were plated in 24-well plates and transfected with the specified plasmids. Following 24 h incubation, the cells were allowed to grow until they reached 95–100% confluence. A linear wound was created in the cell monolayer using a yellow pipette tip, and the cells were then washed twice with PBS (SveviceBio, G0002) to remove any debris. The wound healing process was monitored in 0 h and 24 h using a Cell Imaging Multifunctional Microplate Detection System (BioTek Cytation 5, USA).

## Colony formation assay

The cells were detached using trypsin, counted, and then plated at a density of 500 cells per well in six-well plates, maintained in a 5% $CO_2$ incubator at 37 °C. After a 14-day incubation period, the cells were washed with PBS, fixed in methanol for 20 min at room temperature, and then subjected to staining with 0.1% crystal violet (Aladdin, C110703) for 30 min. Images of the colony formation were captured using a smartphone.

## Transwell assay

Transwell chambers equipped with 8 μm polycarbonate membranes (Corining Inc, 3422) were employed to evaluate in vitro cellular migration and invasion. A total of $2 \times 10^4$ cells suspended in 200 μl

of serum-free medium were seeded in the upper chamber, while $4 \times 10^4$ cells in 200 μl of serum-free medium were placed in the upper chamber pre-coated with BD Matrigel matrix (356234, diluted 1:8 with serum-free medium). The lower chambers received 600 μl of medium supplemented with 10% FBS. Following a 20-hour incubation period, non-migratory and non-invasive cells remaining on the upper membrane surface were removed using a cotton swab, followed by fixation in methanol for 20 min. Subsequently, the membranes were stained with 0.1% crystal violet for 30 min, photographed with a Leica fluorescence stereomicroscope (Nussloch, Germany).

## Cellular immunofluorescence

Cellular immunofluorescence staining was performed as described previously (Yu et al, 2021). Immunofluorescence images were acquired using an Olympus confocal microscope (Tokyo, Japan) and a Nikon confocal microscope (Tokyo, Japan). The following antibodies were used: anti-KI67 antibody (1:100, Proteintech Group, 27309-1-AP), Alexa Fluor® 647 AffiniPure® Goat Anti-Rabbit IgG (1:400, Jackson, #111-605-144), DAPI (1:800, Sigma, 28718-90-3), TRITC-labelled phalloidin (1:100, Thermo Fisher Scientific, R415).

## Immunofluorescence of zebrafish embryos

Zebrafish embryos were fixed with 4% paraformaldehyde (PFA, Sigma, p6148) at 4 °C overnight. The following day, embryos were subjected to a series of increasing methanol concentrations (25%, 50%, 75% and 100%), followed by a 30-minute storage at −20 °C. Embryos were then rehydrated using a series of decreasing methanol concentrations (75%, 50% and 25%). The embryos were subjected to a 3% Triton X-100 treatment for 1 h, followed by a blocking step using PBS containing 10% FBS and 3% Triton X-100 at room temperature for 2 h. Next, the embryos were incubated with primary antibodies at 4 °C overnight. On the following day, the embryos incubated with fluorescent conjugated secondary antibodies and DAPI at room temperature for 2 h. After four washes, the embryos were imaged using an Olympus confocal microscope (Tokyo, Japan) and a Nikon confocal microscope (Tokyo, Japan). The following antibodies were used: anti-KI67 antibody (1:100, Proteintech Group, 27309-1-AP), Alexa Fluor® 594 AffiniPure® Goat Anti-Rabbit IgG (1:400, Jackson, #111-585-046), DAPI (1:800, Sigma, 28718-90-3).

## RNA sequencing (RNA-Seq) analysis and protein-protein interaction network construction

RNA from MDA-MB-231 cells was isolated using TRIZOL. RNA-Seq was performed in the Gene Denovo Biotechnology (Guangzhou, China). Differentially expressed genes (DEGs) were detected using DESeq2. Reactome enrichment analysis and GO enrichment analysis were performed to elucidate the functional implications of DEGs. DEGs with a false discovery rate (FDR) of less than 0.05 were selected for enrichment analysis. The protein-protein interaction (PPI) network involving CDC42 and various guanine nucleotide exchange factors (GEFs) was constructed using the STRING database (https://cn.string-db.org/).

## CDC42-GTP pull-down assay

CDC42 activation was examined using a CDC42 activation assay kit (New East Biosciences, 80701), according to the manufacturer's instructions. MDA-MB-231 cells were lysed and the cell lysates were incubated with an anti-CDC42-GTP antibody (1:1000, New East Biosciences, 26905) at 4 °C for 1.5 h. Then, GTP-bound CDC42 was pulled down by protein A/G agarose (New East Biosciences, 30301) and analyzed by immunoblotting with an anti-CDC42 antibody (1:1000, Proteintech Group, 67212). EasyBlot kit (GeneTex, GTX425858) was used for mitigating background signals attributed to protein A/G.

## Reelin treatment and Reelin function blocking

Prior to the Reelin treatment, MDA-MB-231 cells were subjected to two washes with PBS and then incubated in serum-free DMEM for 6 h. Subsequently, the cells were treated with recombinant human Reelin (RD systems, 8456-MR-050) at a concentration of 100 ng/ml for 16 h.

To block Reelin function, an anti-Reelin monoclonal antibody (Abcam, ab78540) at a concentration of 5 µg/ml was administered to xenografted larvae. This antibody targets the epitope recognized by the function-blocking CR-50 antibody (D'Arcangelo et al, 1997; Ogawa et al, 1995). Additionally, a mouse IgG antibody (ABclonal, AC011) at the same concentration served as a control group for the treatment of zebrafish.

## Molecular docking-based drug screening and Molecular dynamics simulation

The homology model of the LRP8 protein was downloaded from AlphaFold2 (AF-Q14114-F1-v4). Molecular docking was performed to screen out small molecule compounds, from the MCE Bioactive Compound Library Plus database, nicely targeting LRP8 protein. Next, we showed the combination model diagram and docking scores. We also used the molecular dynamics to investigate the protein-ligand binding stability.

## Establishment of nude mice BC xenograft model

The mice experiments had been approved by Experimental Animal Ethics Committee of Guangdong Medical University and the ethical clearance number is AHGDMU-LAC-II(1)-2207-B008. Female nude mice (~8 weeks old) purchased from Beijing Vital River Laboratory Animal Technology Co., Ltd. (Beijing, China), were accommodated in the specific pathogen-free (SPF) animal facility. Mice were anesthetized with tribromoethanol (Aibei Biotechnology, M2960) and MDA-MB-231 (GFP$^+$) cells at a concentration of $5 \times 10^5$ cells per 100 µl were injected into the left cardiac ventricle of mice. The mice were randomly divided into vehicle-treated group and MEN 10207-treated group. The treatment group received intravenous injections of MEN 10207 (3.8 mg/kg) once a day for a duration of 5 days started at 4 days post-injection (dpi), and the vehicle-treated group was treated with saline as a control. By 14 dpi, the mice were sacrificed and the brains were harvested. Next, we used Transparent Embedding Solvent System (TESOS) method with light-sheet fluorescence microscopy system (Nuohai Life Science, China) to acquire three-dimensional images of uniform resolution for whole brains (Yi et al, 2024), and calculated the brain metastatic tumor volume by Imaris software.

## Immunohistochemical staining of mice brain sections

Xenograft model mice of vehicle-treated group at 14 dpi were anesthetized with tribromoethanol and perfused transcardially with saline and 4% PFA. Next, brains were removed, fixed overnight in 4% PFA, equilibrated in 20% sucrose for 24 h and in 30% sucrose for 24 h. Then brains were embedded in OCT and fast frozen in liquid nitrogen. Leica CM1950 cryostat (Nussloch, Germany) was used to cut 40 µm thick sections for immunohistochemical staining. Brain sections were blocked with 1% bovine serum albumin in PBST-T (PBST supplemented with 0.3% Triton X-100) for 1 h at room temperature, followed by an overnight 4 °C incubation with Rabbit anti-CD31 monoclonal antibody (1:200, Abcam, ab222783) diluted in the blocking solution. Then sections were washed with PBST three times and incubated with Alexa Fluor® 594 AffiniPure® Goat Anti-Rabbit IgG (1:500, Jackson, 111-585-046) and DAPI (1:800, Sigma, 28718-90-3) diluted in the blocking solution for 2 h at room temperature. Sections were washed three times again with PBST, mounted with antifade mounting medium and imaged with Nikon confocal microscope (Tokyo, Japan).

## Statistical analysis

Statistical evaluations were conducted utilizing Prism 11 software. The mean ± standard deviation (SD) presented the results. $P$ values for comparisons between two groups were calculated using the

---

### The paper explained

#### Problem

Breast cancer (BC) remains the most prevalent cancer globally, with brain metastasis (BM) being its deadliest complication. While BC cells were traditionally thought to invade the brain by crossing the blood-brain barrier (BBB), recent evidence suggests they can bypass the BBB by migrating along the outer surfaces of blood vessels. However, the molecular drivers of this alternative route and effective therapies targeting it remain unclear, leaving patients with BM few viable treatment avenues.

#### Results

Using a zebrafish xenograft model, we visualized BC cells migrating along cerebral veins in real time, confirming an extravascular pathway to the brain. We identified LRP8, a receptor upregulated in BC patients with BM, as a critical driver of this process. LRP8 promotes cancer cell motility by activating CDC42, a protein that stimulates filopodia formation, enabling cells to "crawl" through the cerebral vasculatures. This mechanism is enhanced by Reelin, a neuronal protein that interacts with LRP8. Targeting this Reelin-LRP8-CDC42 axis with MEN 10207, a neurokinin-2 receptor antagonist, inhibited BC cell migration and reduced BM in zebrafish and mouse models, offering a promising therapeutic strategy.

#### Impact

Our findings redefine how BC invades the brain, highlighting the Reelin-LRP8-CDC42 axis as a key target for preventing BM. The zebrafish model provides a rapid, cost-effective tool for studying metastasis and screening drugs. MEN 10207's efficacy in preclinical models positions it as a potential therapy to block BCBM, addressing an urgent clinical need. By uncovering both a novel metastatic pathway and a candidate treatment, this work opens new avenues for improving outcomes in patients with this devastating complication.

unpaired Student's *t* test (normal distribution) and the Mann–Whitney test (non-normal distribution). Additionally, one-way ANOVA was used to compare values between more than two groups on one dependent variable, while two-way ANOVA was used to test the interaction effects between two factors. Each experiment was replicated and subjected to statistical scrutiny a minimum of three times. A *P* value of less than 0.05 was established as the threshold for statistical significance.

## Data availability

All data necessary for confirming the conclusions are included in this article. The generated RNA-Seq data of MDA-MB-231 cells have been deposited into GEO database: GSE289226. The microarray data of breast cancer patients were acquired from GEO database with accession numbers of "GSE100534" and "GSE52604", respectively.

The source data of this paper are collected in the following database record: biostudies:S-SCDT-10_1038-S44321-025-00260-0.

## Peer review information

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

## Acknowledgements

This work was supported by the National Natural Science Foundation of China (32222028, 32470889), the National Key R&D Program of China (2024YFA1802200), and the high-level talents scientific research start-up funds of the Affiliated Hospital of Guangdong Medical University (2081Z20230015). We thank Yuezhuang Zheng and Qiumei Hong for technical support. We thank Shanshan Ke and Jiali Peng for helping with the three-dimensional reconstruction of images. We also thank Dr. Shouyu Wang from the First Affiliated Hospital of Anhui Medical University, China for excellent technical suggestion.

## Author contributions

**Haofeng Huang**: Data curation; Formal analysis; Investigation; Visualization; Methodology; Writing-original draft. **Min Zhang**: Data curation; Investigation; Visualization; Methodology. **Enyu Huang**: Data curation; Investigation; Visualization; Methodology; Writing-review and editing. **Yongfang Zhao**: Data curation; Supervision; Investigation; Methodology. **Xiaoyu Li**: Data curation; Formal analysis. **Pu Qiu**: Resources; Investigation. **Cairui Li**: Resources; Formal analysis. **Jiahua Tao**: Data curation; Formal analysis; Methodology. **Yuanqi Zhang**: Resources; Methodology. **Lianxiang Luo**: Resources; Methodology. **Guozhu Ning**: Supervision; Methodology. **Ceshi Chen**: Supervision; Methodology. **Jingjing Zhang**: Conceptualization; Resources; Supervision; Funding acquisition; Investigation; Methodology; Project administration; Writing-review and editing.

Source data underlying figure panels in this paper may have individual authorship assigned. Where available, figure panel/source data authorship is listed in the following database record: biostudies:S-SCDT-10_1038-S44321-025-00260-0.

## Disclosure and competing interests statement

The authors declare no competing interests.

# Expanded View Figures

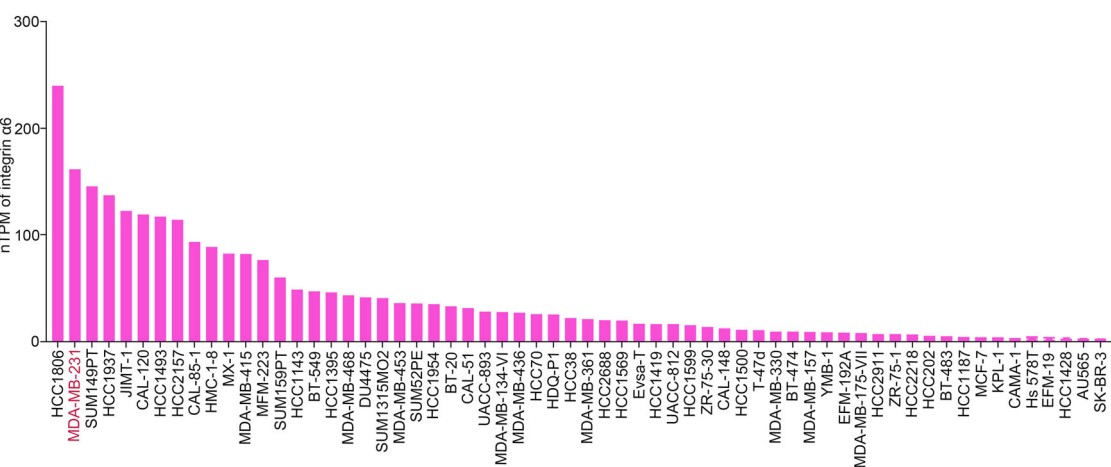

**Figure EV1. Integrin α6 mRNA expression in various human breast cancer cell lines.**

Integrin α6 mRNA expression in various human breast cancer cell lines. Data were obtained from Human Protein Atlas.

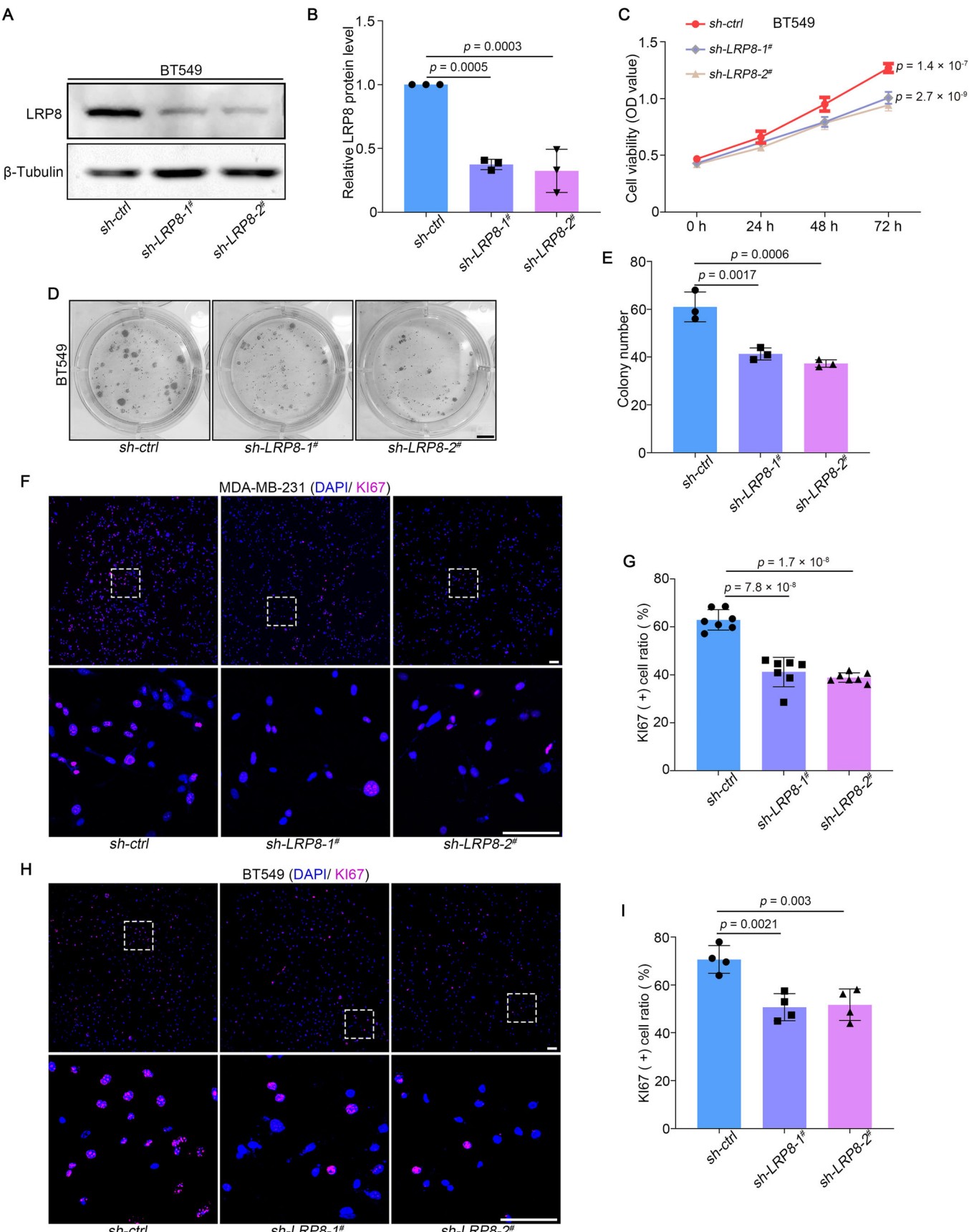

**Figure EV2.** *LRP8* **knockdown inhibits proliferation of TNBC cells.**

(A) BT549 cells transfected with *sh-ctrl*, *sh-LRP8-1#* and *sh-LRP8-2#* were subjected for western blot assay to analyse the protein level of LRP8. (B) Quantitative analysis showed the relative LRP8 protein level normalized to β-tubulin, $n = 3$. (C) Cell viability of BT549 cells transfected with *sh-ctrl*, *sh-LRP8-1#* and *sh-LRP8-2#* was evaluated using the CCK-8 assays, $n = 3$. (D, E) Images and quantification of the number of colonies formed from BT549 cells transfected with *sh-ctrl*, *sh-LRP8-1#* and *sh-LRP8-2#*. Scale bar: 500 μm, $n = 3$. (F) MDA-MB-231 cells transfected with *sh-ctrl*, *sh-LRP8-1#* and *sh-LRP8-2#* were immunostained with KI67 (pure signal) and nuclei were stained by DAPI (blue signal). Scale bar: 100 μm. (G) Quantitative analysis of KI67 positive cell rate in three cell lines of MDA-MB-231 cells in vitro, $n = 7$. $P = 7.8 \times 10^{-8}$ (*sh-ctrl* vs. *sh-LRP8-1#*), $P = 1.7 \times 10^{-8}$ (*sh-ctrl* vs. *sh-LRP8-2#*). (H) BT549 cells transfected with *sh-ctrl*, *sh-LRP8-1#* and *sh-LRP8-2#* were immunostained with KI67 (pure signal) and nuclei were stained by DAPI (blue signal). Scale bar: 100 μm. (I) Quantitative analysis of KI67 positive cell rate in three cell lines of BT549 cells in vitro, $n = 4$. Data information: data are shown as mean ± SD, *P* values were analyzed with one-way ANOVA test (B, E, G, I) and two-way ANOVA test (C). Source data are available online for this figure.

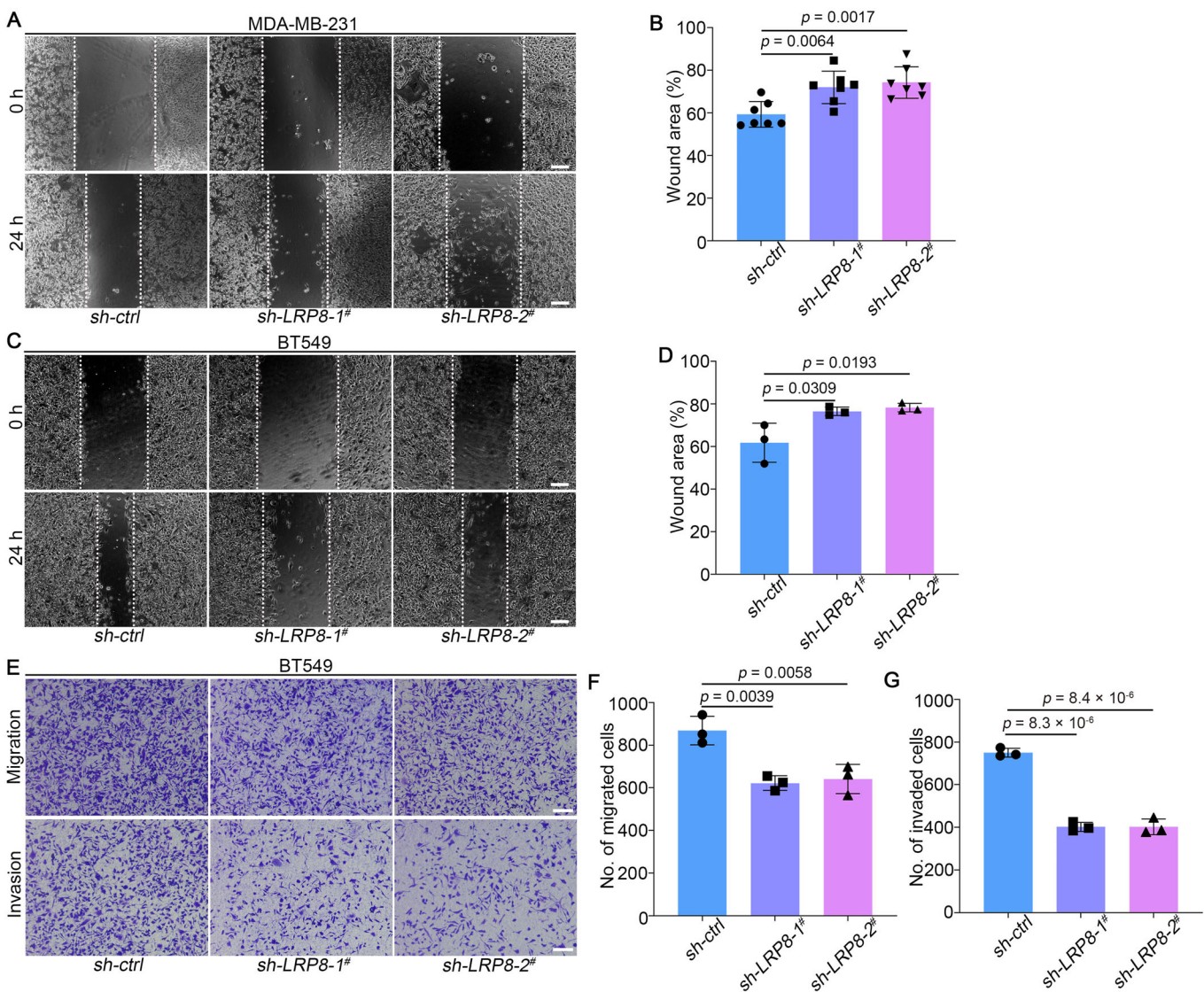

**Figure EV3. LRP8 knockdown inhibits the migration and invasion of TNBC cells.**

(A, B) The investigation of wound healing capabilities in MDA-MB-231 cells transfected with *sh-ctrl*, *sh-LRP8-1*# and *sh-LRP8-2*# were depicted. Assessment of the wound area ratio in 24 h in comparison to baseline measurements at 0 h. Scale bar: 200 μm, $n = 7$. (C, D) Investigation of wound healing capabilities in BT549 cells transfected with *sh-ctrl*, *sh-LRP8-1*# and *sh-LRP8-2*#. Assessment of the wound area ratio in 24 h in comparison to baseline measurements at 0 h. Scale bar: 200 μm, $n = 3$. (E–G) Transwell migration assay and matrigel transwell invasion assay in BT549 cells transfected with *sh-ctrl*, *sh-LRP8-1*# and *sh-LRP8-2*#. Scale bar: 200 μm, $n = 3$. $P = 8.3 \times 10^{-6}$ (*sh-ctrl* vs. *sh-LRP8-1*#), $P = 8.4 \times 10^{-6}$ (*sh-ctrl* vs. *sh-LRP8-2*#). Data information: data are shown as mean ± SD, $P$ values were analyzed with one-way ANOVA test (B, D, F, G). Source data are available online for this figure.

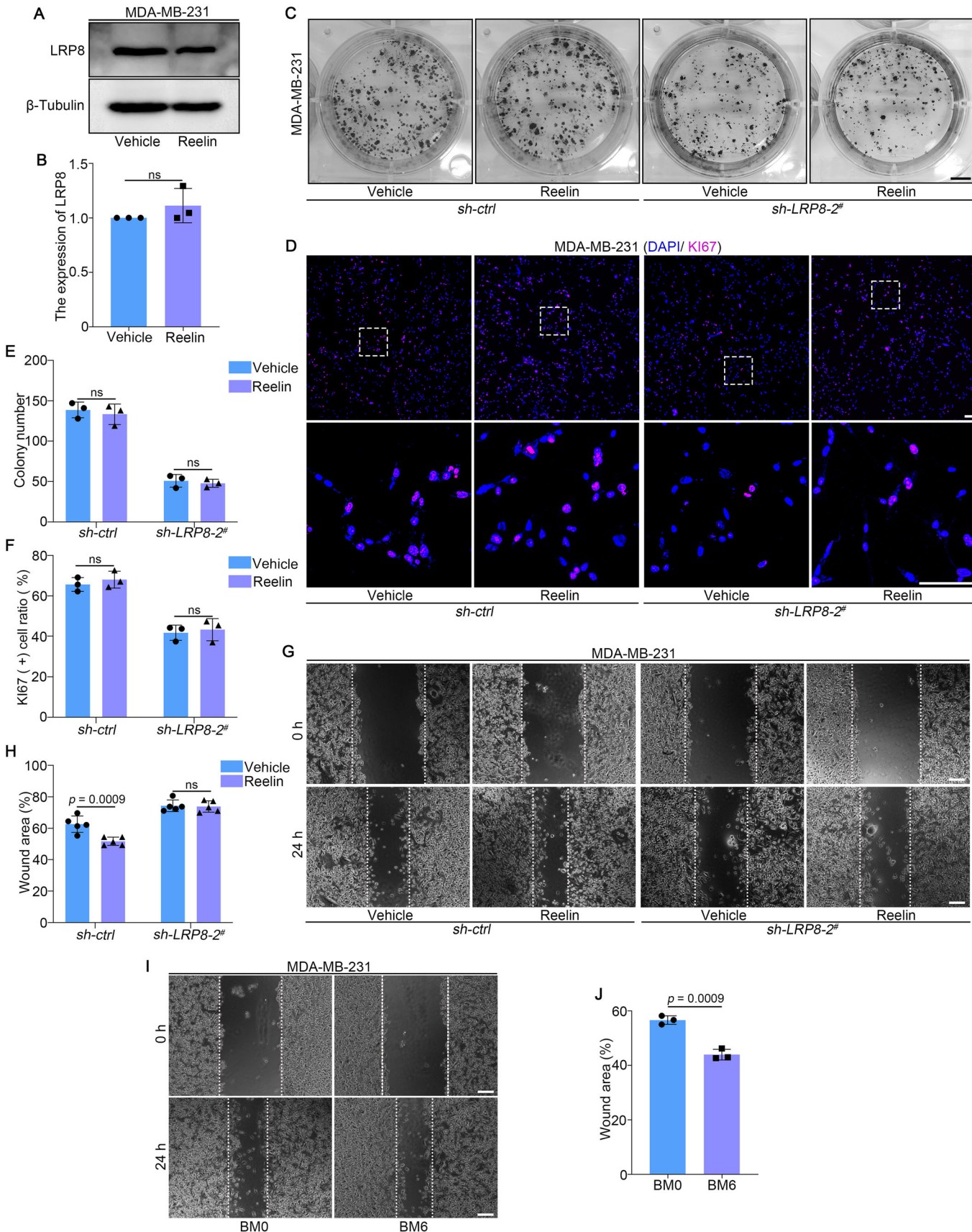

◀ **Figure EV4.  Reelin-LRP8 pathway may not influence the proliferation of MDA-MB-231 cells, but increased the migration.**

(A, B) MDA-MB-231 cells were subjected for western blot assay to analyze the protein level of LRP8 after Reelin treatment. $n = 3$. (C, E) Images and quantification of the number of colonies formed from MDA-MB-231 cells transfected with *sh-ctrl* and *sh-LRP8-2*[#], following Reelin treatment. Scale bar: 500 μm, $n = 3$. (D, F) MDA-MB-231 cells transfected with *sh-ctrl* and *sh-LRP8-2*[#] were immunostained with KI67 (pure signal) and nuclei were stained by DAPI (blue signal) after Reelin treatment. Scale bar: 100 μm, $n = 3$. (G) Wound healing capabilities were investigated in MDA-MB-231 cells transfected with *sh-ctrl* and *sh-LRP8-2*[#] after Reelin treatment. Scale bar: 200 μm. (H) Assessment of the wound area ratio in 24 h in comparison to baseline measurements at 0 h. $n = 5$. (I, J) Investigation and quantitative analysis of wound healing capabilities of BM0 cells and BM6 cells. Scale bar: 200 μm, $n = 3$. Data information: data are shown as mean ± SD, $P$ values were analyzed with two-way ANOVA test (E, F, H) and unpaired Student's $t$ test (B, J). ns non-significant. Source data are available online for this figure.

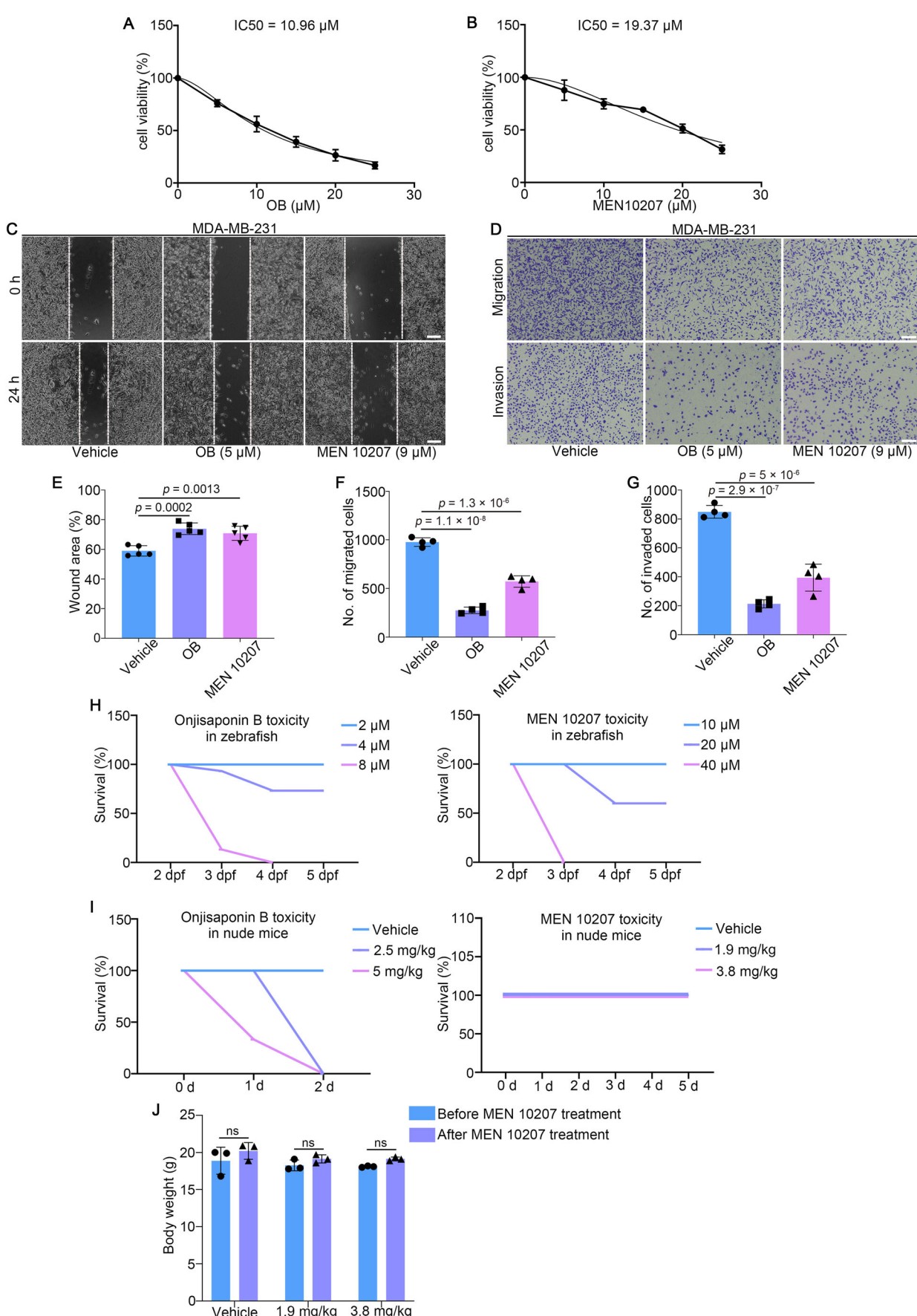

◀ **Figure EV5. OB and MEN 10207 treatment inhibit the migration of MDA-MB-231 cells in vitro and toxicity of OB and MEN 10207 is detected in zebrafish embryos and nude mice.**

(A, B) Effects of OB and MEN 10207 on the cell viability of MDA-MB-231 cells at 24 h, $n = 3$. (C) Investigation of wound healing capabilities in MDA-MB-231 cells treated with vehicle, OB and MEN 10207. Scale bar: 200 μm. (D) Transwell migration assay and matrigel transwell invasion assay in MDA-MB-231 cells treated with vehicle, OB and MEN 10207. Scale bar: 200 μm. (E) Assessment of the wound area ratio in 24 h in comparison to baseline measurements at 0 h. Scale bar: 200 μm, $n = 5$. (F) Quantitative analysis of D showed the migratory abilities. $n = 4$. $P = 1.1 \times 10^{-8}$ (Vehicle vs. OB), $P = 1.3 \times 10^{-6}$ (Vehicle vs. MEN 10207). (G) Quantitative analysis of D showed the invasive abilities. $n = 4$. $P = 2.9 \times 10^{-7}$ (Vehicle vs. OB), $P = 5 \times 10^{-6}$ (Vehicle vs. MEN 10207). (H) Quantification of the proportion of surviving embryos following treatment with OB and MEN 10207 starting at 2 dpi. $n > 10$. (I) Quantification of the proportion of surviving nude mice (approximately 8 weeks old) following treatment with OB and MEN 10207. $n = 3$. (J) Body weight changed in nude mice before and after MEN 10207 treatment. $n = 3$. Data information: data are shown as mean ± SD, $P$ values were analyzed with one-way ANOVA test (E, F, G) and two-way ANOVA test (J). ns non-significant. Source data are available online for this figure.

