## [Peer Review File · EMBO Molecular Medicine]

Reelin-LRP8 signaling mediates brain dissemination of breast cancer cells via abluminal migration

Haofeng Huang, Min Zhang, Enyu Huang, Yongfang Zhao, Xiaoyu Li, Pu Qiu, Cairui Li, Jiahua Tao, Yuanqi Zhang, Lianxiang Luo, Guozhu Ning, Ceshi Chen, and Jingjing Zhang

Corresponding authors: Jingjing Zhang (jingjing.zhang@gdmu.edu.cn) , Guozhu Ning (ningguozhu@gdmu.edu.cn)

Review Timeline:

Submission Date:	11th Feb 25
Editorial Decision:	11th Mar 25
Revision Received:	21st Apr 25
Editorial Decision:	21st May 25
Revision Received:	26th May 25
Accepted:	26th May 25

Editor: Jingyi Hou

Transaction Report:

11th Mar 2025

Dear Jingjing,

Thank you again for submitting your work to EMBO Molecular Medicine. We have now received feedback from two of the three referees who agreed to evaluate your manuscript. Unfortunately, Reviewer #1 was unable to submit a report due to personal reasons. As you will see in the reports below, the other two referees find your study of interest but have raised several concerns that will need to be thoroughly addressed in a major revision of the manuscript.

The referees' recommendations are clear, so I won't repeat the points listed below. It is important to carefully address all the issues raised by the referees. Please feel free to contact me in case you would like to discuss in further detail any of the issues raised by the reviewers.

We would welcome the submission of a revised version within three months for further consideration. As you may already know, our editorial policy allows in principle a single round of major revision, and it is therefore essential to provide responses to the referees' comments that are as complete as possible. EMBO Molecular Medicine has a "scooping protection" policy, whereby similar findings that are published by others during review or revision are not a criterion for rejection. Should you decide to submit a revised version, I do ask that you get in touch after three months if you have not completed it, to update us on the status.

Please also contact us as soon as possible if similar work is published elsewhere. If other work is published, we may not be able to extend the revision period beyond three months.

I look forward to receiving your revised manuscript.

Yours sincerely,
Jingyi

Jingyi Hou
Senior Editor
EMBO Molecular Medicine

We require:

- 1) A .docx formatted version of the manuscript text (including legends for main figures, EV figures and tables). Please make sure that the changes are highlighted to be clearly visible.
- 2) Individual production quality figure files as .eps, .tif, .jpg (one file per figure). For guidance, download the 'Figure Guide PDF': (<https://www.embopress.org/page/journal/17574684/authorguide#figureformat>).
- 3) A .docx formatted letter INCLUDING the reviewers' reports and your detailed point-by-point responses to their comments. As part of the EMBO Press transparent editorial process, the point-by-point response is part of the Review Process File (RPF), which will be published alongside your paper.
- 4) A complete author checklist, which you can download from our author guidelines (<https://www.embopress.org/page/journal/17574684/authorguide#submissionofrevisions>). Please insert information in the checklist that is also reflected in the manuscript. The completed author checklist will also be part of the RPF.

6) It is mandatory to include a 'Data Availability' section after the Materials and Methods. Before submitting your revision, primary datasets produced in this study need to be deposited in an appropriate public database, and the accession numbers and database listed under 'Data Availability'. Please remember to provide a reviewer password if the datasets are not yet public (see <https://www.embopress.org/page/journal/17574684/authorguide#dataavailability>).

12) Author contributions: You will be asked to provide CRediT (Contributor Role Taxonomy) terms in the submission system. These replace a narrative author contribution section in the manuscript.

13) A Conflict of Interest statement should be provided in the main text.

14) Every published paper now includes a 'Synopsis' to further enhance discoverability. Synopses are displayed on the journal webpage and are freely accessible to all readers. They include a short stand first (maximum of 300 characters, including space) as well as 2-5 one-sentences bullet points that summarizes the paper. Please write the bullet points to summarize the key NEW findings. They should be designed to be complementary to the abstract - i.e. not repeat the same text. We encourage inclusion of key acronyms and quantitative information (maximum of 30 words / bullet point). Please use the passive voice. Please attach these in a separate file or send them by email, we will incorporate them accordingly.

Please also suggest a visual abstract to illustrate your article as a PNG file 550 px wide x 300-600 px high.

15) All Materials and Methods need to be described in the main text using our 'Structured Methods' format. According to this format, the Methods section includes a Reagents and Tools Table (listing key reagents, experimental models, software and relevant equipment and including their sources and relevant identifiers) followed by a Methods and Protocols section describing the methods, ideally using a step-by-step protocol format. The aim is to facilitate adoption of the methodologies across labs.

Please download and fill our Reagents and Tools Table template (.docx), which you can find in our author guidelines: <https://www.embopress.org/page/journal/17574684/authorguide#structuredmethods>

When submitting your revised manuscript, please DO NOT include the Reagents and Tools Table in the Methods section of the manuscript but upload it as a separate file choosing the file type "Reagent Table".

***** Reviewer's comments *****

Referee #2 (Comments on Novelty/Model System for Author):

The strength of this paper is showing a new role for LRP8 in Brain metastasis (BM). By characterizing the molecular and cellular basis of this phenotype, the authors found that breast cancer (BC) cells can migrate along the abluminal surface of blood vessels in a zebrafish xenograft model, and LRP8 knockdown significantly inhibited cell proliferation, migration, and invasion both in vitro and in vivo. Guided by previous findings this model can truly monitor the movement of adhesion vessels in brain metastasis of breast cancer cells. Recent studies have found that neural signaling pathways drive brain metastasis of cancer cells. They found that neuronal extracellular matrix protein and Reelin can activate CDC42 through Reelin-LRP8-CDC42 axis, which in turn promotes the migration and invasion of TNBC cells. In molecular docking studies, MEN 10207 inhibits the Reelin-LRP8 signaling pathway. MEN 10207 may play a potential role in the treatment of brain metastasis of breast cancer. Overall, the manuscript features data of high quality and is well written. As detailed below, a few findings should be clarified, and the general interest of this manuscript for readers beyond a specialist community could be increased.

Referee #2 (Remarks for Author):

CONCERNS

- 1) Please clarify why breast cancer cells migrate along the posterior cerebral vein, instead of other blood vessels? Because we can clearly see that the blood vessels of the breast cancer transplant site are relatively rich.
- 2) Fig. 2J, whether overexpression of LRP8 can enhance filopodia formation and cell motility in MCF10a or MCF7?
- 3) Fig. 3E should focus on the areas of cell proliferation and migration, and be appropriately enlarged. Please carefully check the spaces. For example, FIG 3E and 3F, cell proliferation.....
- 4) Please accurately mark the vertical coordinates of Fig. 4H.
- 5) Is there any change in cell morphology of BM6, such as filopodia?
- 6) It is debatable whether or not MEN 10207 could affect cell migration and invasion ability through Reelin-LRP8-CDC42 axis. This notion lacks any direct support from experimental data.
- 7) The sample size is too small in Fig. 6G.
- 8) We cannot determine whether tumors are located in the brain parenchyma rather than blood vessels. To prevent confusion, I recommend brain slice to observe the location of tumor cells.

Referee #3 (Comments on Novelty/Model System for Author):

This model compared to the mammalian model like the mouse model is highly different. It will be good if this model can be justified in the mouse brain metastasis model.

Referee #3 (Remarks for Author):

In this manuscript, Huang et al. applied zebrafish as an in vivo model to investigate the migration and breast tumor formation. Their study revealed Reelin-LRP8-Cdc42 signaling axis as an important pathway to regulate the tumor cell migration into the brain. Using larval zebrafish, the authors could demonstrate real-time cell behavior and migration route which follows a non-canonical abluminal pathway. The questions below should be cleared:

1. For the first time, this study described the dynamic process of the migration of triple-negative breast cancer along the abluminal surface of cerebral blood vessels in an in vivo model, which is similar to the invasion of breast cancer cells along the periphery of the guide vein into meningeal tissue discovered by Whiteley et al. in 2024.
2. The authors identified that LRP8, a signal molecule highly expressed in triple-negative breast cancer, promotes brain metastasis of breast cancer by relying on the neural factor signaling axis Reelin-LRP8.
3. The author screened a Neurokinin-2 receptor antagonist (MEN 10207) which has good effect on inhibiting brain metastasis of breast cancer, by in vitro cell system, in vivo zebrafish and mice models.

This research work once again confirmed that the brain microenvironment plays an important role in brain metastasis of breast cancer. The findings are clearly proved by the experimental evidence.

While I still have some concerns based on the current version of this manuscript:

Major points:

1. In the zebrafish model, is the location of breast cancer cells with brain metastasis in meninges or brain parenchyma?
2. Reelin acts via LRP8 to regulate DAB1 tyrosine phosphorylation. I wonder how about the expression level or phosphorylation level of DAB1 in triple negative breast cancer cells?
3. Was the western blot result validated for the increased LRP8 mRNA level in BM6 cells?
4. Does the small molecule drug MEN 10207 affect the filopodia formation of triple-negative breast cancer cells, and whether the mechanism of MEN 10207 inhibiting brain metastasis of breast cancer is achieved through the LRP8-CDC42 pathway? These results should be demonstrated.
5. Moreover, does MEN 10207 affect the proliferation and apoptosis of triple-negative breast cancer cells? The authors should confirm this by in vitro experiments additionally.
6. The fluorescent signal of cells in Figure 6J is not clear enough, there should be a magnified local fluorescence image.
7. In Figure 6J, green fluorescent cells seem to localize in the surface layer of brain tissue. It is better to detect the localization of green fluorescent cells. Could the authors provide projected sections (deeper inside view) of the brains with tumor cells?
8. I suggest to change the title as is: Reelin-LRP8 signaling mediates brain dissemination of breast cancer cells via abluminal migration. Please do not use the term "metastasis".

Point-by-point Response (EMM-2024-21068)

*****Editorial Comments*****

We require:

Response:

We have prepared the revised manuscript in .docx format, and all figure legends and tables are included. Meanwhile, all the changes and improved contents are marked in red as request.

2) Individual production quality figure files as .eps, .tif, .jpg (one file per figure). For guidance, download the 'Figure Guide PDF':

(<https://www.embopress.org/page/journal/17574684/authorguide#figureformat>).

Response:

We have uploaded the image files in .tif format with high quality under the guidance of the “Figure Guidelines”.

3) A .docx formatted letter INCLUDING the reviewers' reports and your detailed point-by-point responses to their comments. As part of the EMBO Press transparent

editorial process, the point-by-point response is part of the Review Process File (RPF), which will be published alongside your paper.

Response:

We have prepared a response letter with detailed point-by-point responses to address the comments from the editor and both reviewers.

4) A complete author checklist, which you can download from our author guidelines (<https://www.embopress.org/page/journal/17574684/authorguide#submissionofrevisions>). Please insert information in the checklist that is also reflected in the manuscript. The completed author checklist will also be part of the RPF.

Response:

We have prepared the author checklist and uploaded together with the revised manuscript in the submission system.

Response:

The ORCID of both corresponding authors was supplied in the submission system.

6) It is mandatory to include a 'Data Availability' section after the Materials and Methods. Before submitting your revision, primary datasets produced in this study need to be deposited in an appropriate public database, and the accession numbers and database listed under 'Data Availability'. Please remember to provide a reviewer password if the datasets are not yet public (see <https://www.embopress.org/page/journal/17574684/authorguide#dataavailability>). In case you have no data that requires deposition in a public database, please state so in this section. Note that the Data Availability Section is restricted to new primary data

that are part of this study.

Response:

We have uploaded all the primary datasets produced in this study under GEO database. All primary data produced in this study have been uploaded and could be accessed using the supplied link directly. An independent “Data Availability” section was included in the revised manuscript, after the section of “Materials and Methods” (*Line 651-659*).

Response:

We have described in detail the name of the statistical test used to generate error bars and P values, the numbers of independent experiments in each figure legend, including the main figure and EV figure legend.

Response:

We have prepared and compacted the source data according to the request and data list from EMBO source data coordinator (Dr. Hannah Sonntag). A list of the figure

panels for required source data together with the source data were submitted in the submission system.

9) *Our journal encourages inclusion of *data citations in the reference list* to directly cite datasets that were re-used and obtained from public databases. Data citations in the article text are distinct from normal bibliographical citations and should directly link to the database records from which the data can be accessed. In the main text, data citations are formatted as follows: "Data ref: Smith et al, 2001" or "Data ref: NCBI Sequence Read Archive PRJNA342805, 2017". In the Reference list, data citations must be labeled with "[DATASET]". A data reference must provide the database name, accession number/identifiers and a resolvable link to the landing page from which the data can be accessed at the end of the reference. Further instructions are available at <https://www.embopress.org/page/journal/17574684/authorguide#referencesformat>.*

Response:

We thank the Editor for reminding us the inclusion of data citation. In our manuscript, we have described and stated in the “Material and methods” section the public database used in our study, which includes “Databases analysis from Gene Expression Omnibus, UALCAN databases and The Human Protein Atlas” (*Line 410-418*) and “Kaplan-Meier survival curve analysis” (*Line 419-425*). These data were used and applied to generate “EV Fig. 1” and “Fig. 2A-G” respectively.

10) *We replaced Supplementary Information with Expanded View (EV) Figures and Tables that are collapsible/expandable online. A maximum of 5 EV Figures can be typeset. EV Figures should be cited as 'Figure EV1, Figure EV2' etc... in the text and their respective legends should be included in the main text after the legends of regular figures.*

<https://www.embopress.org/page/journal/17574684/authorguide#expandedview>.

Response:

We thank the Editor for the instructions. This manuscript contains 6 main figures and 5 expanded view figures (EV figures) in total, with their legends included in the main text after the legends of regular figures (*Line 982-1049*). Besides, we included 4 expanded view tables (Table EV1-4) in our manuscript, which were provided in a separated file named "EV Tables" (in .xls format), with their legends provided in the same file. Moreover, we have 7 additional movies to support our findings and the legends of each movie are described after the EV figure legends (*Line 1051-1080*).

11) *The paper explained: EMBO Molecular Medicine articles are accompanied by a summary of the articles to emphasize the major findings in the paper and their medical implications for the non-specialist reader. Please provide a draft summary of your article highlighting*

- *the medical issue you are addressing,*
- *the results obtained and*
- *their clinical impact.*

Response:

We have prepared a summary of our story to emphasize the major findings in the paper and their medical implications. The independent file of “The Paper Explained” was submitted together with the main files in the submission system.

12) Author contributions: You will be asked to provide CRediT (Contributor Role Taxonomy) terms in the submission system. These replace a narrative author contribution section in the manuscript.

Response:

We have provided CRediT terms in the submission system. Meanwhile, the content of Author Contributions section was updated to keep consistent with the CRediT terms in the submission system (**Line 661-673**).

13) A Conflict of Interest statement should be provided in the main text.

Response:

The section of “Disclosure and competing interests statement” was provided in the revised manuscript (**Line 690-691**).

14) Every published paper now includes a 'Synopsis' to further enhance discoverability. Synopses are displayed on the journal webpage and are freely accessible to all readers. They include a short stand first (maximum of 300 characters, including space) as well as 2-5 one-sentences bullet points that summarizes the paper. Please write the bullet points to summarize the key NEW findings. They should be designed to be complementary to the abstract - i.e. not repeat the same text. We

encourage inclusion of key acronyms and quantitative information (maximum of 30 words / bullet point). Please use the passive voice. Please attach these in a separate file or send them by email, we will incorporate them accordingly.

Please also suggest a visual abstract to illustrate your article as a PNG file 550 px wide x 300-600 px high.

Response:

We have prepared an independent file of “Synopsis” according to the instructions. Meanwhile, a visual abstract was provided and embedded into the file.

15) All Materials and Methods need to be described in the main text using our 'Structured Methods' format. According to this format, the Methods section includes a Reagents and Tools Table (listing key reagents, experimental models, software and relevant equipment and including their sources and relevant identifiers) followed by a Methods and Protocols section describing the methods, ideally using a step-by-step protocol format. The aim is to facilitate adoption of the methodologies across labs.

Please download and fill our Reagents and Tools Table template (.docx), which you can find in our author guidelines: <https://www.embopress.org/page/journal/17574684/authorguide#structuredmethods>.

When submitting your revised manuscript, please DO NOT include the Reagents and Tools Table in the Methods section of the manuscript but upload it as a separate file choosing the file type "Reagent Table".

Response:

We have provided the updated “Reagents and Tools Table” as an independent file and submitted it in the submission system.

EMBO Molecular Medicine has a "scooping protection" policy, whereby similar

findings that are published by others during review or revision are not a criterion for rejection. Should you decide to submit a revised version, I do ask that you get in touch after three months if you have not completed it, to update us on the status.

Response:

We thank the journal of EMBO Molecular Medicine for the “scooping protection” policy. We have made a revision and submitted our revised manuscript on time.

******* Reviewer's comments *******

Referee #2 (Comments on Novelty/Model System for Author):

The strength of this paper is showing a new role for LRP8 in Brain metastasis (BM). By characterizing the molecular and cellular basis of this phenotype, the authors found that breast cancer (BC) cells can migrate along the abluminal surface of blood vessels in a zebrafish xenograft model, and LRP8 knockdown significantly inhibited cell proliferation, migration, and invasion both in vitro and in vivo. Guided by previous findings this model can truly monitor the movement of adhesion vessels in brain metastasis of breast cancer cells. Recent studies have found that neural signaling pathways drive brain metastasis of cancer cells. They found that neuronal extracellular matrix protein and Reelin can activate CDC42 through Reelin-LRP8-CDC42 axis, which in turn promotes the migration and invasion of TNBC cells. In molecular docking studies, MEN 10207 inhibits the Reelin-LRP8 signaling pathway. MEN 10207 may play a potential role in the treatment of brain metastasis of breast cancer.

Overall, the manuscript features data of high quality and is well written. As detailed below, a few findings should be clarified, and the general interest of this manuscript for readers beyond a specialist community could be increased.

Referee #2 (Remarks for Author):

CONCERNS

1) Please clarify why breast cancer cells migrate along the posterior cerebral vein, instead of other blood vessels? Because we can clearly see that the blood vessels of the breast cancer transplant site are relatively rich.

Response:

We thank the Reviewer for this important comment. According to recent report by Whiteley et al.⁽¹⁾, breast cancer cells can bypass the blood-brain barrier by binding Laminin in vascular basement membranes through integrin $\alpha 6$, a neuronal pathfinding

molecule, and migrate along drainage veins connecting bone marrow to meningeal spaces to enter the meningeal cavity. In our study, database mining revealed high expression levels of Integrin $\alpha 6$ in MDA-MB-231 cells (Fig. EV1), while zebrafish embryonic endothelial cells were also found to abundantly express Laminin⁽²⁾. This may explain why MDA-MB-231 cells preferentially adhere to blood vessels for migration. Regarding the preferential migration along posterior cerebral veins, we attempted to detect Laminin expression in zebrafish posterior cerebral veins (PceVs) using LAMA1 and LAMA4 antibodies for immunofluorescence staining. Unfortunately, it did not work due to antibody incompatibility in zebrafish. Future study using zebrafish-specific Laminin antibodies may reveal the protein expression of Laminins in zebrafish PceVs and could further explain this point.

2) *Fig. 2J, whether overexpression of LRP8 can enhance filopodia formation and cell motility in MCF10a or MCF7?*

Response:

We thank the Reviewer for this critical question. To investigate whether overexpression of LRP8 can enhance filopodia formation and cell motility in MCF10a or MCF7, we transfected MCF7 cells to overexpress LRP8. The upregulated expression level of LRP8 was first validated by Western blotting (Fig. R1A, B). Subsequently, transwell assays demonstrated that LRP8 overexpression significantly enhanced the migration and invasion capabilities of MCF7 cells (Fig. R1C-E). However, immunofluorescence staining experiments revealed that no significant changes in filopodia formation in LRP8-overexpressing MCF7 cells (Fig. R1F-H).

Figure R1. Overexpression of LRP8 can enhance the migratory and invasive abilities in MCF-7 cells. (A) MCF-7 cells transfected with *vector* and *pCMV-LRP8* plasmids were subjected for western blot assay to analyze the protein level of LRP8. (B) Quantitative analysis showed the relative LRP8 protein level normalized to β -tubulin, $n = 4$. (C) Transwell migration assay and matrigel transwell invasion assay in MCF-7 cells transfected with *vector* and *PCMV-LRP8* plasmids. (D, E) Quantitative analysis of (C) showed the migratory and invasive abilities in MCF-7 cells. $n = 3$. (F) MCF-7 cells transfected with *vector* and *PCMV-LRP8* were stained with

TRITC-labeled phalloidin (red) and DAPI (blue) to detect the formation of filopodia. Scale bar: 10 μm . (G, H) Quantification analysis of the number of filopodia per cell and average filopodia length of MCF-7 cells. $n = 5$. Data information: data are shown as mean \pm SD, p -values were analyzed with unpaired Student's t test (B, D, E, G, H). ns: non-significant.

3) Fig. 3E should focus on the areas of cell proliferation and migration, and be appropriately enlarged. Please carefully check the spaces. For example, FIG 3E and 3F, cell proliferation.....

Response:

We thank the Reviewer for pointing out these problems. In the revised manuscript, we enlarged the brain regions of zebrafish embryos (Fig. 3E). It could more clearly demonstrated that MDA-MB-231 cells in the *sh-ctrl* group exhibited significant proliferation and migration in zebrafish compared to the *sh-LRP8-2^{\#}* group (Fig. 3E; Fig. R2). Meanwhile, we have performed a thorough proofreading of the manuscript's text and formatting to avoid any typos or mistakes.

Figure R2. Proliferation of BC cells in zebrafish after transplantation. Representative confocal microscopy images of proliferation of MDA-MB-231 cells in zebrafish after transplantation. MDA-MB-231 cells are shown in green. Scale bar: 200 μ m.

4) *Please accurately mark the vertical coordinates of Fig. 4H.*

Response:

We thank the Reviewer for this critical comment. The vertical coordinates of updated Fig. 4I has been revised for greater precision, with “filopodia per cell” updated to “the average number of filopodia per cell”.

5) *Is there any change in cell morphology of BM6, such as filopodia?*

Response:

We thank the Reviewer for this question. To address this point, we performed the immunofluorescence staining assays on BM0 and BM6 cells. It revealed that no significant changes in filopodia formation in BM6 cells (Fig. R3A-C). However, mRNA expression of downstream effectors in the Reelin-LRP8 pathway, such as *CDC42BPG* and *VAV3*, was upregulated in BM6 cells (Fig. 5L). It is well known that *VAV3* could promote *CDC42* activation, while *CDC42BPG* is a downstream effector of *CDC42*. Previous studies have shown that *CDC42* activation enhances tumor cell migration and invasion⁽³⁾. Thus, the increased migratory and invasive capacities of BM6 cells may be linked to the upregulation of the Reelin-LRP8-*CDC42* signaling axis.

Figure R3. The formation of filopodia in BM0 cells and BM6 cells. (A) BM0 and BM6 cells were stained with TRITC-labelled phalloidin (red) and DAPI (blue) to detect the formation of filopodia. Scale bar: 10 μm . (B, C) Quantification analysis of the number of filopodia per cell and average filopodia length of BM0 and BM6 cells. $n = 6$. Data information: data are shown as mean \pm SD, p -values were analyzed with unpaired Student's t test (B, C). ns: non-significant.

6) *It is debatable whether or not MEN 10207 could affect cell migration and invasion ability through Reelin-LRP8-CDC42 axis. This notion lacks any direct support from experimental data.*

Response:

We thank the Reviewer for this critical question. By performing pull-down assays following drug treatment, it demonstrated that MEN 10207 suppressed intrinsic CDC42 activation in MDA-MB-231 cells and inhibited the Reelin-mediated CDC42 activation pathway (Fig. R4A, B).

Figure R4. The change of GTP-CDC42 level following MEN 10207 treatment. (A) MDA-MB-231 cells treated with vehicle and MEN 10207 were subjected to western blot assay to analyze the protein level of activated GTP-bound CDC42, following Reelin treatment. (B) Quantitative analysis revealed the relative GTP-bound CDC42 level normalized to total CDC42 protein. $n = 5$. Data information: data are shown as mean \pm SD, p -values were analyzed with one-way ANOVA test (B). ns: non-significant.

7) The sample size is too small in Fig. 6G.

Response:

We thank the Reviewer for pointing out this important question. During the revision, we have performed additional experiments to enlarge the zebrafish sample sizes to 10 embryos. The results demonstrated that both OB and MEN 107 could significantly inhibit the proliferation and migration of MDA-MB-231 cells in zebrafish brain (Fig. 6D-F; Fig. R5A, B).

Figure R5. The effects of OB and MEN 10207 treatment in zebrafish xenograft models. (A) Representative confocal microscopy images of zebrafish xenograft models treated with vehicle, OB and MEN 10207. Scale bar: 100 μ m. (B) Quantitative analysis of E showed the relative fluorescence intensity of GFP *in vivo*. The measurement of fluorescence of GFP at 24 hpi was used as the baseline. n = 10. (C) Quantification analysis of the migrated distance of brain metastatic cells along PceV in zebrafish at 72 hpi was conducted. n = 10. Data information: data are shown as mean \pm SD, *p*-values were analyzed with one-way ANOVA test (B, C). ns: non-significant.

8) We cannot determine whether tumors are located in the brain parenchyma rather than blood vessels. To prevent confusion, I recommend brain slice to observe the location of tumor cells.

Response:

We thank the Reviewer for raising this critical concern. In the zebrafish breast cancer cell xenograft model, 3D imaging and reconstruction confirmed that migrating tumor cells traverse along the abluminal surface of blood vessels (Fig. R6A; Movies 1-3). In the murine breast cancer cell xenograft model, brains harvested from mice at 14 days post-engraftment underwent sectioning, immunofluorescence staining, confocal imaging, and 3D sectional analysis, revealing that brain-colonizing tumor cells predominantly localized to the extravascular brain parenchyma surrounding cerebral vasculature (Fig. 6J; Fig. R6B).

Figure R6. The localization of MDA-MB-231 cells within zebrafish and mice brain. (A) Three-dimensional images of lateral view (Top) and dorsal view (Bottom) to visualize the position of MDA-MB-231 cells in zebrafish brain. The MDA-MB-231 cells are shown in green and blood vessels are shown in red. Skin stained by Alexa Fluor™ 647 cadaverine, is shown in purple. Bottom: Scale bar: 50 μ m. (B) Immunofluorescence staining (scale bar: 100 μ m) for CD31 (red signal) and DAPI (blue signal) was conducted on mice brain section and three-dimensional imaging (scale bar: 50 μ m) was performed to visualize the interaction between MDA-MB-231 cells and the blood vessels in mouse brain. MDA-MB-231 cells are displayed in green.

Referee #3 (Comments on Novelty/Model System for Author):

This model compared to the mammalian model like the mouse model is highly different. It will be good if this model can be justified in the mouse brain metastasis model.

Response:

We thank the Reviewer for the recognition of our work. Briefly, to validate our observations in mammalian models, brains from mice at 14 days post-engraftment were harvested, sectioned, and subjected to immunofluorescence staining. Subsequent confocal imaging and 3D sectional acquisition confirmed that brain-colonizing tumor cells predominantly localized to the extravascular brain parenchyma surrounding cerebral vasculature. Furthermore, intravenous administration of MEN 10207 post-engraftment significantly suppressed tumor cell colonization in the brain. These critical findings demonstrate robust consistency between zebrafish and murine breast cancer cell xenograft models.

Referee #3 (Remarks for Author):

In this manuscript, Huang et al. applied zebrafish as an in vivo model to investigate the migration and breast tumor formation. Their study revealed Reelin-LRP8-Cdc42 signaling axis as an important pathway to regulate the tumor cell migration into the brain. Using larval zebrafish, the authors could demonstrate real-time cell behavior and migration route which follows a non-canonical abluminal pathway. The questions below should be cleared:

1. For the first time, this study described the dynamic process of the migration of triple-negative breast cancer along the abluminal surface of cerebral blood vessels in an in vivo model, which is similar to the invasion of breast cancer cells along the periphery of the guide vein into meningeal tissue discovered by Whiteley et al. in 2024.

2. The authors identified that LRP8, a signal molecule highly expressed in triple-negative breast cancer, promotes brain metastasis of breast cancer by relying on the neural factor signaling axis Reelin-LRP8.

3. The author screened a Neurokinin-2 receptor antagonist (MEN 10207) which has good effect on inhibiting brain metastasis of breast cancer, by in vitro cell system, in vivo zebrafish and mice models.

This research work once again confirmed that the brain microenvironment plays an important role in brain metastasis of breast cancer. The findings are clearly proved by the experimental evidence. While I still have some concerns based on the current version of this manuscript:

Major points:

1. In the zebrafish model, is the location of breast cancer cells with brain metastasis in meninges or brain parenchyma?

Response:

We thank the Reviewer for this critical question. The location of breast cancer cells with brain metastasis is in brain parenchyma. To show this in detail and in higher magnifications, we stained the transplanted zebrafish with Alexa Fluor™ 647 Cadaverine to label the skin first. The confocal imaging and 3D reconstruction revealed that tumor cells migrating along the posterior cerebral veins (PceV) predominantly colonized deep parenchymal regions of the zebrafish brain (Fig. R7A, B; Movies 1-3).

Figure R7. The localization of MDA-MB-231 cells within PceV of zebrafish. (A)

Three-dimensional images of dorsal view to visualize the position of MDA-MB-231 cells in zebrafish brain. The MDA-MB-231 cells are shown in green and blood vessels are shown in red. Skin stained by Alexa Fluor™ 647 Cadaverine, is shown in purple. Scale bar: 50 μm. (B) Three-dimensional reconstruction was performed in dorsal view to visualize the position of MDA-MB-231 cells in zebrafish brain. The MDA-MB-231 cells are shown in green, blood vessels are shown in pink. Skin stained by Alexa Fluor™ 647 Cadaverine, is shown in purple. Scale bar: 50 μm.

2. *Reelin acts via LRP8 to regulate DAB1 tyrosine phosphorylation. I wonder how about the expression level or phosphorylation level of DAB1 in triple negative breast cancer cells?*

Response:

We thank the Reviewer for raising this concern. Western blotting analysis of four cell lines revealed that DAB1 and phosphorylated DAB1 (P-DAB1) exhibited significantly higher expression levels in triple-negative breast cancer (TNBC) cell lines (BT549 and MDA-MB-231) compared to mammary epithelial cells (MCF10A) and luminal-type breast cancer cells (MCF7).

Figure R8. The phosphorylation level of DAB1 in breast epithelial cell line and breast cancer cell lines. (A) Western blot analysis was performed on MCF-10A, MCF-7, BT549, and MDA-MB-231 cell lines to evaluate phosphorylation of DAB1 level. (B) Quantitative analysis

revealed the relative phosphorylation level of DAB1 normalized to total DAB. $n = 3$. Data information: data are shown as mean \pm SD, p -values were analyzed with one-way ANOVA test (B). ns: non-significant.

3. Was the western blot result validated for the increased LRP8 mRNA level in BM6 cells?

Response:

Western blotting analysis of LRP8 in BM0 and BM6 cells revealed no significant differences of its expression between these two cell lines (Fig. R9A, B). However, mRNA expression of downstream effectors in the Reelin-LRP8 pathway, such as *CDC42BPG* and *VAV3*, was upregulated in BM6 cells (Fig. 5L). It is well known that *VAV3* promotes *CDC42* activation, while *CDC42BPG* functions as a downstream effector of *CDC42*. Published studies indicate that *CDC42* activation enhances tumor cell migration and invasion.⁽³⁾ Thus, the heightened migratory and invasive capacities of BM6 cells may correlate with upregulation of the Reelin-LRP8-*CDC42* signaling axis.

Figure R9. The LRP8 protein level of brain metastatic cells. (A) Western blot analysis was performed on BM0, BM3 and BM6 cells to evaluate LRP8 level. (B) Quantitative analysis revealed the relative LRP8 protein level normalized to β -tubulin. $n = 3$. Data information: data are shown as mean \pm SD, p -values were analyzed with one-way ANOVA test (B). ns: non-significant.

4. Does the small molecule drug MEN 10207 affect the filopodia formation of triple-negative breast cancer cells, and whether the mechanism of MEN 10207 inhibiting brain metastasis of breast cancer is achieved through the LRP8-CDC42 pathway? These results should be demonstrated.

Response:

We thank the Reviewer for these critical comments. Pharmacological treatment with MEN 10207 and pull-down assays demonstrated that MEN 10207 inhibited both intrinsic CDC42 activation and Reelin-mediated CDC42 activation in MDA-MB-231 cells (Fig. R10A, B). Cellular immunofluorescence assays further revealed that MEN 10207 suppressed filopodia formation in these cells (Fig. R10C-E). These findings confirm that MEN 10207 suppresses breast cancer brain metastasis by targeted inhibition of the Reelin-LRP8 signaling axis.

Figure R10. The MEN 10207 treated effects on MDA-MB-231 cells. (A) MDA-MB-231 cells treated with vehicle and MEN 10207 were subjected to western blot assay to analyze the protein level of activated GTP-bound CDC42, following Reelin treatment. (B) Quantitative analysis revealed the relative GTP-bound CDC42 level normalized to total CDC42 protein. n = 5. (C) MDA-MB-231 cell treated with vehicle and MEN 10207 were stained with TRITC-labeled phalloidin (red signal) and DAPI (blue signal) to detect the formation of filopodia. Scale bar: 10 μm . (D, E) Quantification analysis of the number of filopodia per cell and average filopodia length of MDA-MB-231 cells. n = 9. Data information: data are shown as mean \pm SD, *p*-values were analyzed with one-way ANOVA test (B) and unpaired Student's t test (D, E). ns: non-significant.

5. Moreover, does MEN 10207 affect the proliferation and apoptosis of triple-negative breast cancer cells? The authors should confirm this by in vitro experiments additionally.

Response:

By detection of cell proliferation and apoptosis, we found that MEN 10207 treatment for 24 hours significantly inhibited MDA-MB-231 cell proliferation (Fig. R11A, B). Although apoptosis increased statistically, the proportion of apoptotic cells remained low (<5%), indicating minimal impact on overall apoptosis in MDA-MB-231 cells (Fig. R11C, D).

Figure R11. The proliferation and apoptosis of MDA-MB-231 cell following the MEN 10207 treatment. (A) MDA-MB-231 cells treated with vehicle and MEN 10207 were immunostained with KI67 (pure signal) and DAPI (blue signal). Scale bar: 100 μm. (B) MDA-MB-231 cells treated with vehicle and MEN 10207 were immunostained with Caspase-3 (pure signal) and DAPI (blue signal). Scale bar: 100 μm. (C) Quantitative analysis of KI67 positive cell rate in two groups of MDA-MB-231 cells *in vitro*, n = 8. (D) Quantitative analysis of Caspase-3 positive cell rate in two groups of MDA-MB-231 cells *in vitro*, n = 6. Data information: data are shown as mean ± SD, *p*-values were analyzed with unpaired Student's t test (B, D).

6. The fluorescent signal of cells in Figure 6J is not clear enough, there should be a magnified local fluorescence image.

Response:

We thank the Reviewer for this suggestion. The magnified images clearly revealed the

distribution of tumor cells within the murine brain (Fig. R12). We have included the improved new images in the revised manuscript (Fig. 6H).

Figure R12. The brain dissemination of MDA-MB-231 cells following the MEN 10207 treatment. Representative Light-sheet fluorescence microscopy images of whole brains of nude mice xenograft models treated with vehicle, and MEN 10207. Scale bar: 1000 μ m.

7. In Figure 6J, green fluorescent cells seem to localize in the surface layer of brain tissue. It is better to detect the localization of green fluorescent cells. Could the authors provide projected sections (deeper inside view) of the brains with tumor cells?

Response:

We thank the Reviewer for this comment. Using Imaris software, we performed 360° horizontal rotation of the entire mouse brain (Movie 5) and X-Z plane cross-sectional analysis, which revealed that green fluorescent cells were diffusely distributed within the brain parenchyma rather than on the external surface (Fig. R13).

Figure R13. The X-Z section of whole brains of nude mice xenograft models. Representative light-sheet fluorescence microscopy images of whole brains of nude mice xenograft models treated with vehicle, and MEN 10207. MDA-MB-231 cells are shown in green. Scale bar: 1000 µm. (B) The X-Z section of MDA-MB-231 cells in the brain. MDA-MB-231 cells are shown in green. Scale bar: 1000 µm.

8. I suggest changing the title as is: *Reelin-LRP8 signaling mediates brain dissemination of breast cancer cells via abluminal migration. Please do not use the term "metastasis".*

Response:

We thank the Reviewer for this critical suggestion. We totally agree with the Reviewer to revise the term “brain metastasis” to “brain dissemination” in the manuscript title and text. In accordance with this terminology adjustment, the term “brain metastasis” will be retained in sections describing the murine xenograft model (where established metastases are histologically confirmed), while “brain dissemination” will be used in

zebrafish xenograft model sections to reflect the observed process of tumor cell migration from the transplantation site to the brain region.

Reference

- (1) Whiteley AE, et al. Breast cancer exploits neural signaling pathways for bone-to-meninges metastasis. *Science*. 2024.
- (2) Eve AMJ, Smith JC. Knockdown of Laminin gamma-3 (Lamc3) impairs motoneuron guidance in the zebrafish embryo. *Wellcome Open Res*. 2017.
- (3) Zhang X, et al. ARHGEF37 overexpression promotes extravasation and metastasis of hepatocellular carcinoma via directly activating Cdc42. *J Exp Clin Cancer Res*. 2022.

21st May 2025

Dear Jingjing,

Thank you for submitting your revised manuscript to EMBO Molecular Medicine. We have now received the enclosed report from the two referees who re-assessed your work. As you will see, the referees are now supportive, and I am pleased to inform you that we will be able to accept your manuscript pending the following amendments:

1. Please address the remaining minor comment from Reviewer #3.

On a more editorial level, please do the following:

1. Reduce the number of keywords to five.

2. EV figures should be uploaded as individual figure files, one figure per file.

3. Please combine the funding information with the Acknowledgement section.

4. Remove the Authors' Contributions section from the manuscript file.

5. The primary corresponding author's institutional email address should be added to the manuscript title page.

6. Please download and fill our Reagents and Tools Table template (.docx), which you can find in our author guidelines: <https://www.embopress.org/page/journal/17574684/authorguide#structuredmethods>

When submitting your revised manuscript, please DO NOT include the Reagents and Tools Table in the Methods section of the manuscript but upload it as a separate file choosing the file type "Reagent Table".

7. Please move "the paper explained" section into the main manuscript.

8. Materials & Methods should be renamed to "Methods". "Ethics approval and consent to participate" should be incorporated into the relevant experimental descriptions within the "Methods" section.

9. Data availability: please remove the reviewer token and ensure that these datasets will be made publicly available upon acceptance.

10. EV tables: In the legends, the nomenclature needs to be corrected to Table EV1 -Table EV4.

11. EV Movies: remove the corresponding legends from the manuscript text file. Zip the legends together with the corresponding movie files and update the nomenclature to Movie EV1, Movie EV2, etc.

12. Please update the order of manuscript sections as follows- The Paper Explained, Introduction, Results, Discussion, Methods, Acknowledgements, Disclosure and competing interests statement, References, Figure legends, Tables and their legends, Expanded View Figure legends.

13. Please address the following issues related to figure legends:

- Please note that the exact p values are not provided in the legends of figures 2F, G; 3C, J-L; 4E, G, I, J; 5E, G, I, Q; 6E, F; EV2G; EV3G; EV5 F, G.

- Please indicate the statistical test used for data analysis in the legends of figures 2F, G.

- Please note that the box plots need to be defined in terms of minima, maxima, centre, bounds of box and whiskers, and percentile in the legends of figures 2F, G; 5J, K

- Please note that information related to n is missing in the legends of figures 5J, K; EV5 A, B

- Please note that the white arrow heads are not defined in the legend of figure 1E, 3M, 4H, 5P, 6D, J. This needs to be rectified.

Please submit your revised manuscript within two weeks. I look forward to seeing a revised form of your manuscript as soon as possible.

Kind regards,
Jingyi

Jingyi Hou
Senior Editor
EMBO Molecular Medicine

*** Instructions to submit your revised manuscript ***

***** Reviewer's comments *****

Referee #2 (Comments on Novelty/Model System for Author):

N/A

Referee #2 (Remarks for Author):

The authors have address all my previous concerns. I have no more questions.

Referee #3 (Comments on Novelty/Model System for Author):

Breast cancer brain metastasis is an important issue. The findings in this report deepen the mechanistic understanding of breast cancer metastasis to the brain.

Referee #3 (Remarks for Author):

Other breast brain metastatic models shall be discussed and compared in the discussion section.

******* Reviewer's comments *********Referee #3 (Remarks for Author):**

Other breast brain metastatic models shall be discussed and compared in the discussion section.

Response:

We thank the Reviewer for this important comment. In the Discussion section of our revised manuscript, we have discussed and compared several BCBM xenograft models reported in recent studies to the zebrafish BCBM xenograft model applied in this study. The detailed text is as “Immunodeficient murine xenograft models serve as indispensable tools in breast cancer metastasis research, particularly for investigating brain metastatic mechanisms (Fernando *et al*, 2022). Current methodological advances demonstrate that BCBM models can be established through multiple approaches, for example: (1) Gan *et al*. successfully developed a brain metastasis model by intracardiac injection of brain-tropic metastatic TNBC cells into female mice (Gan *et al*, 2024); (2) Whiteley *et al*. revealed through EO771-tdT cell engraftment in C57BL/6 mice that breast cancer cells circumvent the blood-brain barrier via migration along the abluminal surface of emissary veins to reach the leptomeningeal space (Whiteley *et al*, 2024); (3) Clinical translation models using patient-derived TNBC cells have been established through intracisternal or intracarotid arterial administration, while BT474 cell injection via the intracarotid artery provides another validated modeling strategy (Cordero *et al*, 2022; Kitamura *et al*, 2021). However, traditional xenograft models require extended durations, substantial financial investments, and pose significant challenges for monitoring dynamic *in vivo* progression (Gamble *et al*, 2021; Stoletov *et al*, 2007) (Line 322 to 334)” .

References

Cordero A, Ramsey MD, Kanojia D, Fares J, Petrosyan E, Schwartz CW, Burga R, Zhang P, Rashidi A, Castro B *et al* (2022) Combination of tucatinib and neural stem cells secreting anti-HER2 antibody prolongs survival of mice with metastatic brain cancer. *Proc Natl Acad Sci U S A* 119

Fernando W, Coyle KM, Marcato P (2022) Breast Cancer Xenograft Murine Models. *Methods Mol Biol* 2508: 31-44

Gamble JT, Elson DJ, Greenwood JA, Tanguay RL, Kolluri SK (2021) The Zebrafish Xenograft Models for Investigating Cancer and Cancer Therapeutics. *Biology (Basel)* 10

Gan S, Macalinao DG, Shahoei SH, Tian L, Jin X, Basnet H, Bibby C, Muller JT, Atri P, Seffar E *et al* (2024) Distinct tumor architectures and microenvironments for the initiation of breast cancer metastasis in the brain. *Cancer Cell* 42: 1693-1712.e1624

Kitamura Y, Kanaya N, Moleirinho S, Du W, Reinshagen C, Attia N, Bronisz A, Revai Lechtich E, Sasaki H, Mora JL *et al* (2021) Anti-EGFR VHH-armed death receptor ligand-engineered allogeneic stem cells have therapeutic efficacy in diverse brain metastatic breast cancers. *Sci Adv* 7

Stoletov K, Montel V, Lester RD, Gonias SL, Klemke R (2007) High-resolution imaging of the dynamic tumor cell vascular interface in transparent zebrafish. *Proc Natl Acad Sci U S A* 104: 17406-17411

Whiteley AE, Ma D, Wang L, Yu SY, Yin C, Price TT, Simon BG, Xu KR, Marsh KA, Brockman ML *et al* (2024) Breast cancer exploits neural signaling pathways for bone-to-meninges metastasis. *Science* 384: eadh5548

26th May 2025

Dear Jingjing,

Congratulations on an excellent manuscript, I am pleased to inform you that your manuscript has been accepted for publication in the EMBO Molecular Medicine. Thank you for your comprehensive response to referee concerns. It has been a pleasure to work with you to get this to the acceptance stage.

Kind regards,
Jingyi

Jingyi Hou
Senior Editor
EMBO Molecular Medicine
